# Regulatory mechanisms of incomplete huntingtin mRNA splicing

Andreas Neueder[1,2], Anaelle A. Dumas[1], Agnesska C. Benjamin [1] & Gillian P. Bates [1]

Huntington's disease is caused by a CAG repeat expansion in exon 1 of the *HTT* gene. We have previously shown that exon 1 *HTT* does not always splice to exon 2 producing a small transcript (*HTTexon1*) that encodes the highly pathogenic exon 1 HTT protein. The mechanisms by which this incomplete splicing occurs are unknown. Here, we have generated a minigene system that recapitulates the CAG repeat-length dependence of *HTTexon1* production, and has allowed us to define the regions of intron 1 necessary for incomplete splicing. We show that manipulation of the expression levels of the splicing factor SRSF6, predicted to bind CAG repeats, modulates this aberrant splicing event and also demonstrate that RNA polymerase II transcription speed regulates the levels of *HTTexon1* production. Understanding the mechanisms by which this pathogenic exon 1 HTT is generated may provide the basis for the development of strategies to prevent its production.

[1] UCL Huntington's Disease Centre, Department of Neurodegenerative Disease and Dementia Research Institute, UCL Institute of Neurology, University College London, London WC1N 3BG, UK. [2] Present address: Department of Neurology, Ulm University, Ulm 89081, Germany. Correspondence and requests for materials should be addressed to A.N. (email: andreas.neueder@uni-ulm.de) or to G.P.B. (email: gillian.bates@ucl.ac.uk)

Huntington's disease (HD) is a devastating hereditary neurodegenerative disorder that manifests with motoric, cognitive and psychiatric symptoms[1]. The mutation is a CAG repeat expansion in exon 1 of the huntingtin gene (HTT)[2], which translates to a polyglutamine (polyQ) track in the huntingtin protein (HTT). In addition to the three full length isoforms[3–5], other isoforms consisting of a combination of the inclusion of cryptic exons[6,7], retention of parts of introns and alternative splice site usages[7–9] have been identified. However, these isoforms are subject to nonsense mediated RNA decay and will have no functional consequences. In contrast, the highly pathogenic exon 1 HTT protein with an expanded polyQ track is sufficient to model numerous HD related phenotypes in many species[10–12]. We have shown that exon 1 HTT is produced by a block in splicing of HTT exon 1 to exon 2 that generates a small transcript, comprising the 5′ UTR, exon 1 and terminating at cryptic polyA sites in intron 1 (HTTexon1)[13]. This transcript is produced in a CAG repeat length-dependent manner in all knock-in mouse models of HD and in HD patient tissue[14]. We have proposed that this block in splicing is at least partly mediated by abnormal binding of the splicing factor SRSF6 to the CAG repeat. This in turn could interfere with the formation of the spliceosome at the 5′ splice site and/or expose cryptic polyA sites in intron 1[13,15].

The production of HTTexon1 by incomplete splicing is CAG repeat-length dependent[13,14]. Therefore, somatic instability of CAG repeats, which can lead to vastly increased CAG repeat lengths, would in turn lead to much higher levels of HTTexon1 and of the pathogenic exon 1 HTT protein[14]. Recent genome-wide association studies uncovered several DNA fidelity maintenance factors that influence HD onset and/or progression[16,17] and are also known to modulate CAG repeat stability[18–20]. This is also influenced by transcription elongation by RNA polymerase II (PolII), in particular certain chromatin marks and R-loops, stable DNA:RNA hybrid structures, correlate with an increase in repeat instability[21,22]. R-loops also play an important role in the regulation of alternative splicing[23] and the termination of transcription[24]. Transcription and splicing are tightly linked and influence each other[25]. The 'window of opportunity' model of splicing states that a certain kinetic window of transcription will allow splicing with a defined pattern. In other words, fast elongation speeds usually favor exclusion and slow elongation speeds inclusion of exons with weaker 5′ and/or 3′ splicing signals[25]. Additionally, polyA site selection and protein binding to the nascent RNA, influenced by differential folding of the RNA due to different PolII speeds, might be tied to transcriptional speeds[25]. Fitting with this hypothesis, and further strengthening the link between transcription and splicing, PolII pauses at the 3′ and 5′ splice sites, which define the exons[26,27] and there is a polyA site dependent checkpoint of transcription termination[28].

To facilitate the analysis of mechanisms that contribute to the incomplete splicing of HTT, we generated a human cell-based minigene system. In this model, we observe the same CAG repeat length-dependent splicing of Htt minigenes that we discovered in mouse models of HD and in HD patients[13,14]. Using this system, we can define the minimal sequences necessary for canonical splicing of Htt, as well as the intronic regions that are required to induce incomplete splicing. Furthermore, we show that modulation of the splicing factor SRSF6 regulates exon 1 to exon 2 splicing in the context of an expanded CAG repeat, highlighting the important function of SRSF6 in this process. Given the tight link between transcription and splicing, we analyzed PolII transcription across the minigenes. We find clear differences in PolII occupancy along the minigenes that contain control as compared to elongated CAG repeats. Corroborating these findings, treatment with drugs that affect transcription and introducing 'road blocks' in the gene by CRISPR/dCAS result in differences in the levels of HTTexon1 production.

## Results

**Generation of a cell model of incomplete splicing of Htt.** To dissect the mechanism underlying incomplete splicing in HD, we constructed a variety of mouse Htt minigenes. We chose to express these in a human cell line (human embryonic kidney fibroblasts—HEK293) (Fig. 1a) for a number of reasons. First, the mouse Htt sequences could be distinguished from endogenous human HTT. Second, one of the cryptic polyA sites utilized in the human gene is located 7.3 kb into intron 1, limiting the feasibility of cloning multiple comparative constructs, because the human HTT intron 1 sequences are extremely GC rich and difficult to work with[14]. Finally, the similarity of the 5′ region of the mouse and human HTT genes, and the high level of conservation between the human and mouse splicing machinery suggested that this would be a feasible approach

All constructs contained the mouse Htt promoter, 5′ UTR, exon 1, the first (5′) 917 bp of intron 1, the last (3′) 338 bp of intron 1, exon 2 fused to a FLAG-tag and mouse Htt 3′ UTR sequences including the first mouse polyA site (Fig. 1a). We hypothesized that these domains would constitute the core sequences required for the splicing of the minigenes (Fig. 1b, short). To determine, which additional intronic sequences would be required to recapitulate incomplete splicing, we inserted additional segments of intron 1 downstream to the core 5′ intronic sequences (Fig. 1b). These included the sequences that were found by RNAseq[13] to be part of HTTexon1 (Fig. 1b, medium), or, in addition, the following 1.3 kb, which included GT/GA repeats and a predicted terminator like hairpin, both possibly acting as transcriptional modulators (Fig. 1b, long). As a control, for a situation in which no exon 1 to exon 2 splicing had occurred, we created minigenes that consisted of the mouse Htt promoter, 5′ UTR, exon 1 and the first (5′) 917 bp of intron 1 only (Fig. 1b, ex1 only). Constructs were generated with a range of CAG repeat lengths including: $(CAG)_7$, $(CAG)_{23}$, $(CAG)_{32}$, $(CAG)_{41}$, $(CAG)_{54}$ and $(CAG)_{103}$.

Constructs were transfected into HEK293 cells using the 'Flp-In' system to create an isogenic series of stable lines. The generation of HTTexon1 involves the usage of cryptic polyA sites in intron 1, of which several are predicted for mouse Htt (Supplementary Fig. 1A). We had previously identified the site at 677 bp into intron 1 as being used in the HdhQ150 mouse model. This site is present in all of the minigene constructs. A second site was predicted by the RNAseq reads from the HdhQ150 mice, that extended up to approximately 1.2 kb into intron 1[13] and we can now confirm that this is located 1145 bp into intron 1 (Supplementary Fig. 1B). We applied 3′RACE to the minigene lines and found that only the first cryptic polyA site was used (Fig. 1c and Supplementary Fig. 1C), to produce a polyadenylated transcript (HTTexon1) in the long intron lines, but not in the short and medium intron lines (Fig. 1c).

In the control situation with $(CAG)_7$, splicing should produce an exon 1-exon 2-FLAG fusion protein. With an expanded CAG repeat, the Htt minigene should be at least partially incompletely spliced and an exon 1 HTT protein should also be produced (Fig. 1d). Consistent with this hypothesis, we could detect the properly spliced exon 1-exon 2-FLAG fusion proteins and an exon 1 HTT protein fragment in the long intron minigene lines with an expanded CAG repeat consistent with the 3′RACE data (Fig. 1e and Supplementary Fig. 2, compare Fig. 1c).

We next used real-time quantitative PCR (qPCR) to identify the minigene transcripts that were present in the cell lines. As outlined above, canonical splicing with $(CAG)_7$, should lead to

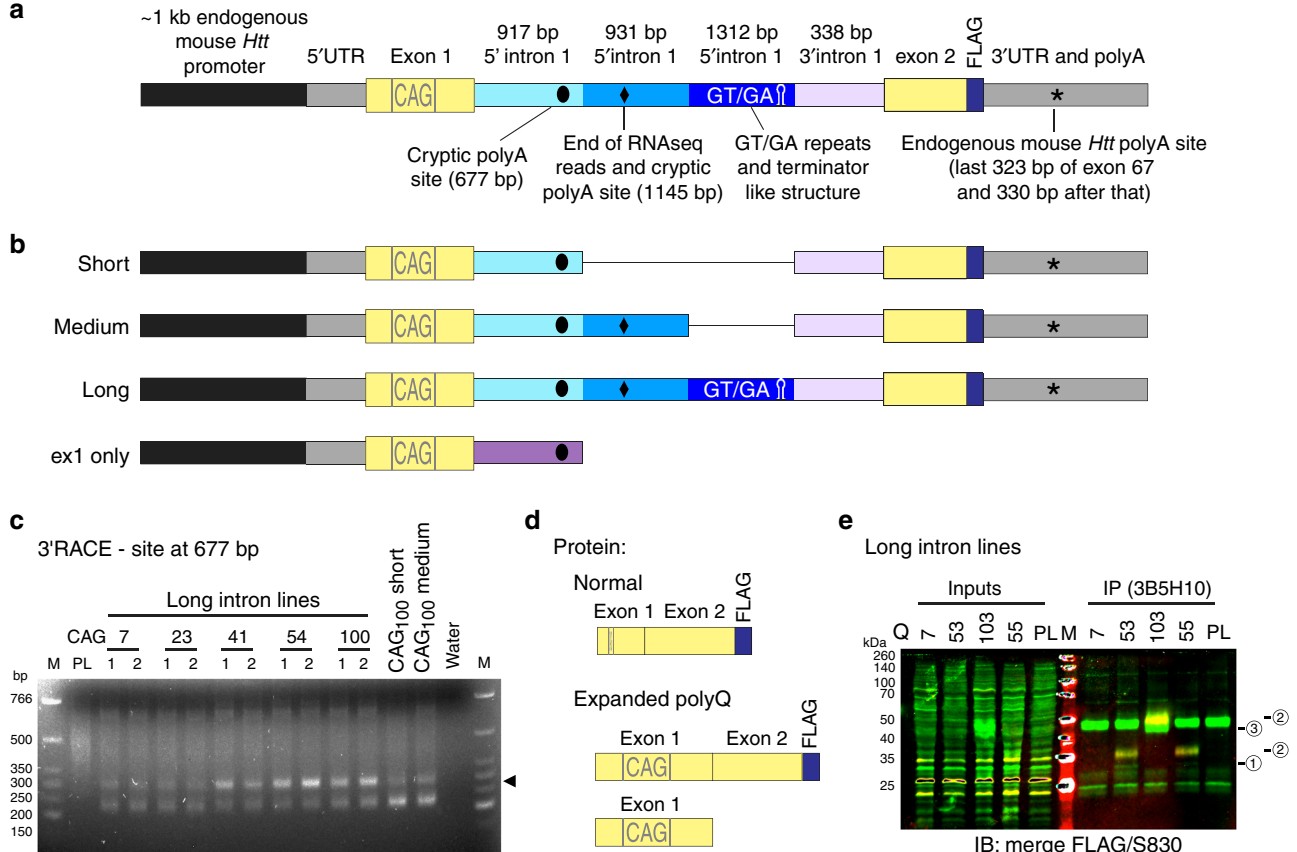

**Fig. 1** Development of a cell based system that recapitulates incomplete splicing of *Htt*. **a** Schematic showing the elements of the *Htt* minigenes. **b** Three minigene constructs differed only in the length of 5′ intron sequences (short: 917 bp; medium: 1848 bp; long: 3160 bp). One construct contained only the coding sequence for exon 1 *HTT* and the first 916 bp of intron 1 (ex1 only). **c** 3′RACE analysis showed that the cryptic polyA site at 677 bp into intron 1 (arrowhead) was only used in the long minigene lines with a threshold of about 40 CAGs. **d** Proteins that will be expressed from these constructs: Splicing will generate an exon 1-exon 2-FLAG fusion protein (control and expanded CAG). In the case of an expanded CAG, an exon 1 HTT protein will also be generated due to incomplete splicing of exon 1 to exon 2. **e** Overlay image of spliced (FLAG) and exon 1 HTT (S830) containing fragments. HTT fragments were immunoprecipitated (IP) with 3B5H10 coupled magnetic beads and immunoprobed (IB) with antibodies as indicated (please see Supplementary Fig. 2 for complete protein analysis). FLAG-tag detects the properly spliced exon 1–exon 2 fragment (2) (see also Fig. 1d). The S830 antibody recognized Q50 and Q100 containing spliced (2), as well as incompletely spliced fragments (1: Q50, 3: Q100). Q7 containing fragments were not detected because they are not efficiently immunoprecipitated with the anti-polyQ antibody 3B5H10. PL parent line; M marker

the generation of a properly spliced exon 1-exon 2 transcript (Fig. 2a). For longer CAG repeats (>(CAG)$_{40}$) a subset of the splicing reactions should be blocked and *HTTexon1* should be generated (Fig. 2a, right panel). *HTTexon1* contains sequences of the 5′ part of intron 1 (light blue in Fig. 2a), which are normally spliced out and degraded. We used the retention of these sequences as a surrogate measure for incomplete splicing (Fig. 2b, intron 1 assay). Canonical splicing was measured with a qPCR assay, in which the forward primer binds in exon 1 and the reverse primer binds in exon 2 of the minigene constructs (Fig. 2b, spliced exon 1-exon 2 assay).

We then examined the extent to which splicing had occurred in the minigene transcripts. To exclude the potential influence of different levels of transcription when comparing minigene derived exonic or intronic transcripts between constructs, they were first normalized to their respective 5′ UTR levels, unless otherwise noted, as these should not be influenced by any downstream events (see also Fig. 2b, 5′ UTR assay). The level of the exon 1 to exon 2 fusion transcript was measured to determine the extent to which complete splicing had occurred. In all lines, we could detect properly spliced minigene transcripts for all CAG repeat lengths (Figs. 2d, f, h and j). The short intron lines showed the highest, and the long intron lines the lowest, expression levels

over housekeeping (HK) genes (Fig. 2j). Previous publications have shown a reduction in full-length mutant *HTT* levels that occurs in mouse models of HD and in patient tissue[13,14]. We only observed a decrease in spliced transcript levels with increasing CAG repeat length in the long intron lines (Fig. 2h).

Incomplete splicing results in the retention of 5′ intronic sequences in the *HTTexon1* transcript, and we used these levels as a measure for incomplete splicing (Fig. 2a, b). Interestingly, the short intron lines, which gave rise to properly spliced minigene transcripts (Fig. 2d) and thus contain all sequences necessary for canonical splicing, showed a minor CAG repeat length-dependent retention of intron 1 (Fig. 2e). Also the addition of subsequent intronic sequences (medium lines) did not result in a CAG repeat dependent increase in incomplete splicing of the minigene constructs (Fig. 2g). In contrast, the long intron minigene lines exhibited readily detectable levels of intronic sequences (Fig. 2i). Consistent with mouse data, incomplete splicing increased with increasing CAG repeat length, and at longer repeats, ≥(CAG)$_{54}$, reached very high levels, comparable to the expression levels of the exon 1 only construct (Fig. 2i, compare long intron to exon 1 only). Furthermore, intron 1 levels (Fig. 2k) were largely inversely correlated to spliced levels (Fig. 2j), indicating that the incompletely spliced transcripts originated from a common and

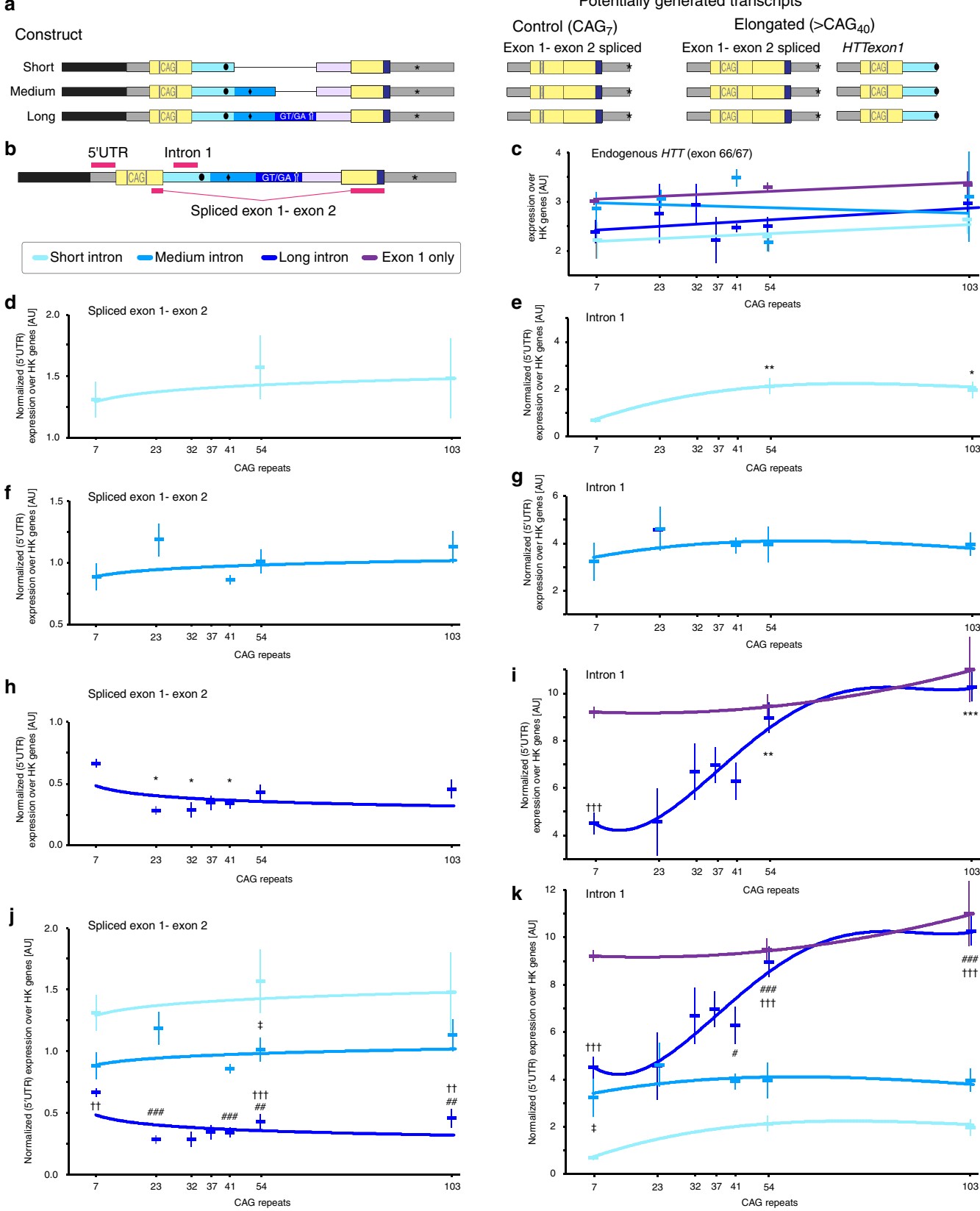

**Fig. 2** Incomplete splicing of *Htt* in the minigene lines. **a** Schematic showing possible transcripts arising from the different minigene constructs. For a control CAG repeat (CAG)$_7$ only properly spliced exon 1–exon 2 transcripts are expected. In the case of an elongated CAG repeat (>(CAG)$_{40}$), in addition to the canonically spliced transcripts, *HTTexon1* is potentially generated. **b** Position of the qPCR assays. *HTTexon1* generation is measured by the amount of retained intron 1 (intron 1 assay). **c–k** Transcript analysis in minigene expressing cell lines. Data are mean ± s.e.m.; $n \geq 3$ independent cell lines/CAG-length/intron-length; two-way ANOVA with Bonferroni post hoc. Comparison of CAG-length to (CAG)$_7$ for each of the respective construct: *$p < 0.05$, **$p < 0.01$, ***$p < 0.001$. Comparison of long vs. medium intron constructs for a given CAG-length: #$p < 0.05$, ##$p < 0.01$, ###$p < 0.001$. Comparison of long vs. short intron constructs for a given CAG-length: ††$p < 0.01$, †††$p < 0.001$. Comparison of medium vs. short intron constructs for a given CAG-length: ‡$p < 0.05$. **c** Endogenous *HTT* levels (parent line) were not changed due to the expression of any of the minigene constructs with any CAG-length. **d**, **f**, **h** Canonically spliced exon 1–exon 2 transcripts. A change with longer CAG lengths was only visible in the long intron lines (**h**). **e**, **g**, **i** Incomplete splicing of the minigene constructs measured by retention of 5′ intron 1 sequences. Significantly CAG repeat length dependent generation of *HTTexon1* was only detected in the long intron lines starting at a threshold of about 40 CAGs (**i**). **j**, **k** Spliced exon 1–exon 2 minigene transcripts (**j**) were largely inversely correlated with incompletely spliced minigenes (**k**)

endonucleolytically cleaved pre-mRNA and not through additional transcription. To determine whether expression of the minigenes might interfere with the expression of endogenous full-length human *HTT* in trans, we measured the levels of exon 66/67 containing *HTT* transcripts by qPCR (Fig. 2c). There were no consistent CAG repeat length-dependent, or construct dependent, differences in endogenous *HTT* levels.

Since CAG repeats are unstable, we analyzed the CAG repeat lengths in our cell lines during continuous growth over 15 passages (approximately 50 days). We only detected instability in the (CAG)$_{100}$ lines, which was independent of the occurrence of incomplete splicing (Supplementary Fig. 3A and 3B, long vs. medium intron 1 lines). The levels of incompletely spliced minigenes remained unchanged (Supplementary Fig. 3C), while endogenous *HTT* levels declined (Supplementary Fig. 3D).

In summary, we have defined the essential regions of the 5′ *Htt* gene necessary for canonical splicing, as well as for the incomplete splicing of *Htt*. Furthermore, our cell model showed CAG repeat length-dependent incomplete splicing and used the same cryptic polyA site as in mouse models of HD.

**SRSF6 levels modulate the amount of *HTTexon1* production**. We have previously shown that the general splicing factor SRSF6 binds to the 5′ end of *Htt* transcripts, consistent with the prediction that it recognizes a CAG repeat[13]. We propose that SRSF6 at this ectopic location could interfere with spliceosome assembly at the 5′ splice site of intron 1 and/or inhibit the protection of cryptic polyA sites in intron 1 by U1 snRNP (U1 RNA containing small nuclear ribonucleoproteins). We increased (Fig. 3a–e) and reduced (Fig. 3f–i) the levels of SRSF6 to test whether this could modulate the amount of incompletely spliced *Htt*. Since we observed pronounced levels of incompletely spliced *HTTexon1* in cell lines with a highly expanded CAG repeat, (CAG)$_{100}$ in comparison to (CAG)$_7$, we focused on these two CAG repeat lengths, in the long intron 1 lines, for all further experiments. Overexpression of mouse *Srsf6* or human *SRSF6* led to highly increased transcript (Fig. 3a, b) and SRSF6 protein levels (Fig. 3c) independent of the CAG repeat length. Although there was no detectable reduction in the levels of the spliced minigenes when either mouse *Srsf6* or human *SRSF6* were overexpressed, (Fig. 3d), the amount of incomplete splicing in the (CAG)$_{100}$ line was increased with overexpression of mouse *Srsf6* and even more so for human *SRSF6* (Fig. 3e). This discrepancy may be because mouse SRSF6 does not fully integrate into the human spliceosome, due to a 5 amino acid difference at the C-terminus (97% overall amino acid homology). siRNA mediated knock-down of *SRSF6* led to significantly decreased transcript (Fig. 3f) and protein (Fig. 3g) levels for both CAG repeat lengths. Unexpectedly, we observed that the basal *SRSF6* transcript, but not SRSF6 protein levels were lower in the CAG$_{100}$ compared to the (CAG)$_7$ lines (Figs. 3f, g, scramble). We did not detect any difference in

the levels of spliced minigenes when *SRSF6* was knocked-down (Fig. 3h). In contrast, the reduction in *SRSF6* levels resulted in significantly lower levels of incomplete splicing in the CAG$_{100}$ line (Fig. 3i).

Taken together, we could show that, consistent with our aforementioned hypothesis, higher levels of SRSF6 increased the levels of incomplete splicing, and lower levels of SRSF6 led to significantly lower levels of *HTTexon1* generation.

In order to determine whether the SRSF6 modulatory effects on *Htt* splicing could be recapitulated with other splicing factors, we overexpressed SRSF1, SRSF2, or SRSF3 in the long intron minigene lines (Supplementary Fig. 4). In contrast to the stimulatory effect on incomplete splicing when SRSF6 was overexpressed (Fig. 3), the overexpression of each of these three SR proteins led to a decrease in incompletely spliced *Htt* minigenes, while normal splicing was not changed (Supplementary Fig. 4). The overexpression of these splicing factors most likely influences the splicing of the *Htt* minigenes through secondary effects, based on changes in the transcriptome.

To determine whether the splicing machinery was dysregulated in general, we used the KEGG pathway genes for spliceosome (mouse: mmu03040; human: hsa03040) and analyzed their expression levels in several tissues of a mouse model of HD at 6 months of age, and in human post mortem brain tissue (Supplementary Fig. 5). There was no general dysregulation of the splicing machinery, and no consistent changes between tissues. Given that we have previously shown that incomplete splicing of *HTT* occurs in these HD mice at 2 months of age[13] and in this post mortem brain region[14], this incomplete splicing is likely to be caused by a transcript specific mechanism, rather than to general dysregulation of the splicing machinery.

**The position of the CAG repeat influences splicing of *Htt***. We next tested whether the presence of a CAG repeat would be sufficient to induce splicing changes, irrespective of its' spatial or genetic context. The binding of SRSF6 to the CAG repeat could interfere with spliceosome formation at the 5′ splice site (end of exon 1). Therefore, moving the CAG repeat away from the 5′ splice site could potentially reduce this interference effect. To test this, we created minigenes, in which a (CAG)$_{23}$ or (CAG)$_{100}$ repeat flanked by 19 bp upstream and 18 bp downstream sequences was placed either in the 5′ UTR (BamHI site) or in intron 1 (KpnI site) of the CAG$_7$ containing long intron backbone (Fig. 4a). The 5′ UTR integrated lines retained a certain level of incompletely spliced *HTTexon1*, albeit to a lower extent than the normal minigenes, and a concomitant decrease in the spliced exon 1–exon 2 transcript (Fig. 4b). In contrast, integration of the CAG repeats in intron 1 did not induce incomplete splicing (Fig. 4c). However, we could not detect any spliced product in these lines (Fig. 4c). We hypothesized that the integration into the 5′ region of intron 1 might have interfered with the assembly of

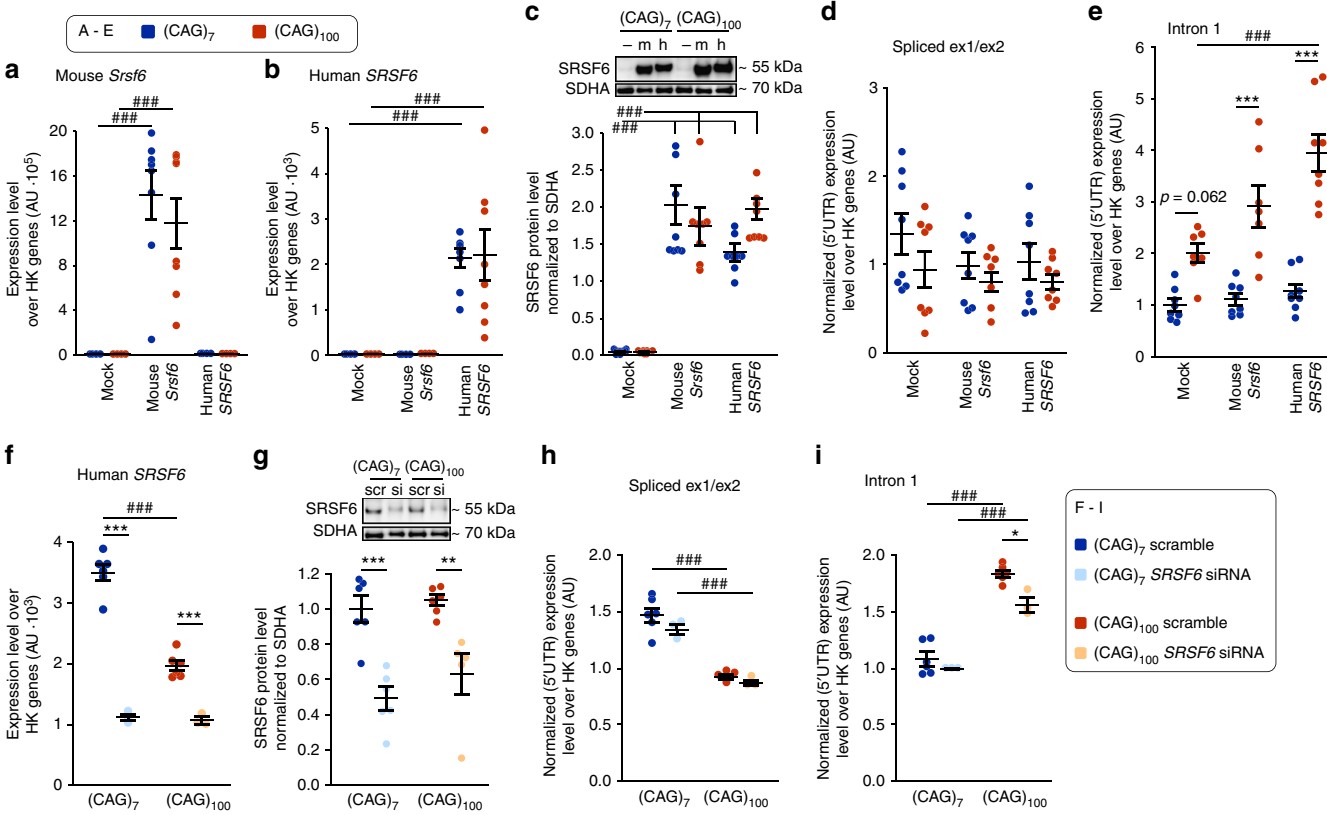

**Fig. 3** SRSF6 levels modulate incomplete splicing of *Htt*. **a–e** Overexpression of *SRSF6* increases the amount of incomplete splicing. Mouse *Srsf6* (**a**) and human *SRSF6* (**b**), respectively, were over-expressed to the same extent in the CAG$_7$ and CAG$_{100}$ cell lines. **c** Western blot data confirmed the over-expression of mouse and human SRSF6 for both CAG repeat lengths. - = mock; m = mouse SRSF6; h = human SRSF6. Uncropped blots can be found in Supplementary Fig. 8A. **d** Spliced exon 1–exon 2 transcripts were not significantly changed due to the overexpression of mouse or human *SRSF6*. **e** Intron 1 containing transcripts were increased to statistically significantly levels when human *SRSF6* was over-expressed. Individual data points and the mean ± s.e. m. are shown. $n$ = 8 independent experiments/CAG-length; two-way ANOVA with Tukey post hoc. Effect of SRSF6 overexpression for a given CAG-length: $^{###}p < 0.001$. Effects due to CAG-length for a given treatment: $^{***}p < 0.001$. **f–i** Knock-down of *SRSF6* by siRNA treatment (s12740, ThermoFisher) decreased the amount of incomplete splicing. *SRSF6* levels were decreased by siRNA treatment on transcript (**f**) and protein (**g**) levels for both CAG repeat lengths. scr = scramble; si = siRNA treatment. Uncropped blots can be found in Supplementary Fig. 8B. **h** Spliced exon 1–exon 2 transcripts were not changed due to the knock-down of *SRSF6*. **i** There was a statistically significant reduction in the amount of incompletely spliced minigene in the CAG$_{100}$ line. Individual data points and the mean ± s.e.m. are shown. $n$ = 3–6 independent experiments/CAG-length; two-way ANOVA with Tukey post-hoc. Treatment for a given CAG-length: $^*p < 0.05$, $^{**}p < 0.01$, $^{***}p < 0.001$. Treatment x CAG-length $^{###}p < 0.001$

the spliceosome at the 5′ splice site or with site selection, which would lead to larger spliced transcripts that might be difficult to detect by qPCR. To test this, we amplified exon 1 to exon 2 spanning sequences with the same primers as in the exon 1–exon 2 spliced qPCR assay (Fig. 4d). We detected a single band for the (CAG)$_{23}$ line in which the CAG repeat had been integrated into the 5′UTR (Fig. 4d, BamHI, 1), representing a properly spliced exon 1–exon 2 mRNA. In the lines in which the CAG repeats had been integrated into intron 1, we could only detect this product at very low levels in the (CAG)$_{23}$ line, together with additional multiple longer PCR products (Fig. 4d, KpnI CAG$_{23}$, 2). There was no properly spliced product detectable in the KpnI (CAG)$_{100}$ lines (Fig. 4d), confirming the qPCR result (Fig. 4c). We sequenced some of the additional larger transcripts (Fig. 4d, 3-7). Comparison of all 5′ splice sites with the consensus sequence (Fig. 4e) confirmed that the 5′ splice site of *Htt* in the smallest product corresponded to the exon 1–exon 2 spliced transcript (Figs. 4d–f, 1 and 2). We uncovered several novel 5′ splice sites that had become activated upon integration of the CAG repeat into intron 1 (Fig. 4f).

To analyze if a CAG repeat alone would be sufficient to interfere with splicing in another genetic context, we introduced a (CAG)$_{23}$ or (CAG)$_{100}$ repeat into unrelated *SDHA* or *SPP1* minigenes (Supplementary Fig. 6) but were unable to detect changes in their splicing pattern.

In summary, a CAG repeat alone is not sufficient to alter splicing, but needs to be in a precise spatial position to the 5′ splice site.

**PolII elongation rate modulates the amount of *HTTexon1*.** Transcriptional speed, i.e., elongation rates of PolII, and splicing are tightly intertwined[25]. Therefore, we set out to analyze whether transcriptional speeds varied along the minigenes with different CAG repeat lengths and could be contributing to the level of incomplete splicing. We used an antibody against the C-terminal domain (CTD) of PolII, which recognizes phosphorylated serine 2 in the YSPTSPS CTD repeat and is associated with actively elongating PolII, to immunoprecipitate PolII/chromatin complexes (ChIP). We immunoprecipitated higher levels of PolII associated minigene sequences from all along the constructs in the (CAG)$_{100}$ versus the (CAG)$_7$ lines (Fig. 5a). This difference was statistically significant for sequences immediately after promoter escape (5′ UTR) and in the 3′ region of the 5′ intronic

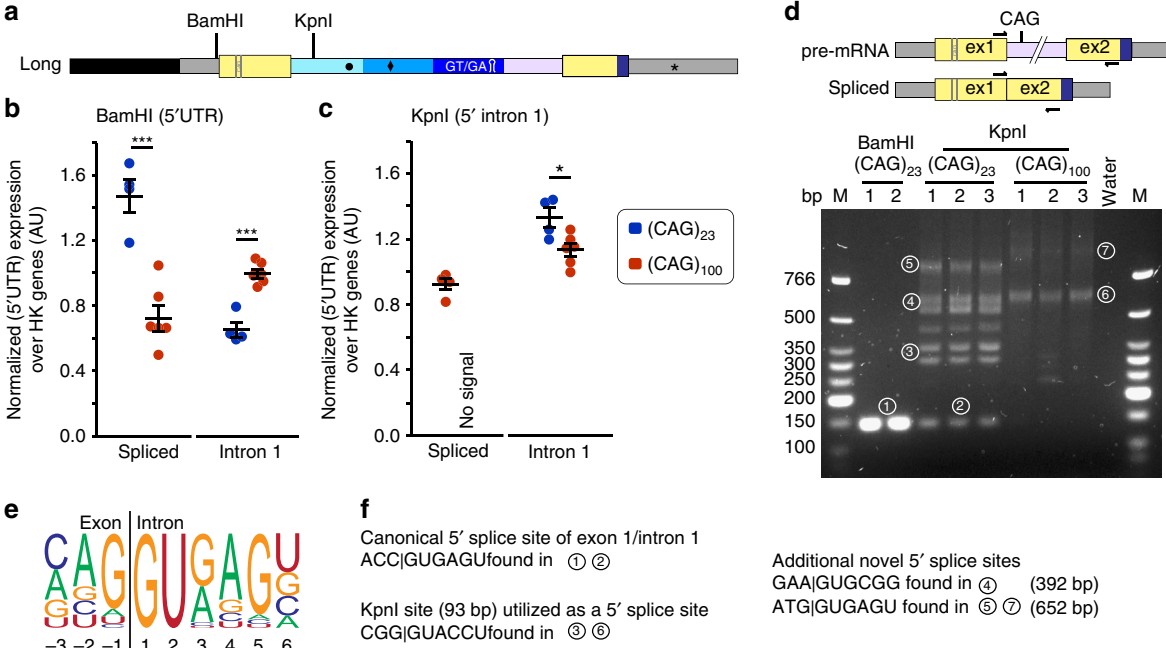

**Fig. 4** The spatial relation of the CAG repeats to the 5′ splice site influences incomplete splicing of *Htt*. **a** Schematic showing the integration sites for the CAG repeat in the (CAG)₇ long minigene backbone. Transcript analysis of BamHI (**b**) and KpnI (**c**) integrated (CAG)₂₃ and (CAG)₁₀₀ repeats. Individual data points and the mean ± s.e.m. are shown. $n \geq 4$ independent cell lines/integration site, one-way ANOVA with Bonferroni post hoc. CAG-length: $*p < 0.05$, $***p < 0.001$. **d** RT-PCR with the same primers as used in the exon 1–exon 2-spliced qPCR assay (see Supplementary Table 2). In addition to the canonically spliced product (1), the integration of a CAG repeat at the KpnI site led to the appearance of multiple RT-PCR products (2-7). **e** The consensus 5′ splice site sequence from -3 to + 6 bp. **f** Novel 5′ splice sites that were utilized when the CAG repeat was inserted into the KpnI site. In bands 3 and 6: the 3′ KpnI site, used to integrate the CAG repeats, functions as a 5′ splice site. In bands 4, 5 and 7: sequences further downstream of the integrated CAGs in intron 1, resembling the consensus 5′ splice site became utilized

sequences (at about 3 kb from the exon 1/intron 1 junction). Transcript levels of 5′ UTR containing minigene sequences were only slightly higher in the CAG₁₀₀ compared to the CAG₇ lines (Fig. 5b, about 1.2-fold). In contrast the PolII co-precipitated 5′ UTR fragments were about 1.7-fold higher (Fig. 5a, $p = 0.016$). If we infer rate of transcription from PolII occupancy, for both CAG repeat lengths, our data suggest that PolII holoenzymes slow down after transcribing through the repeat and speed up again towards the end of intron 1 (Fig. 5a). We also observed a significantly higher occupancy of PolII after the GT/GA repeats and the predicted terminator like hairpin in the (CAG)₁₀₀ line (Fig. 5a, 2.7–3.1 kb), indicating a slower transcriptional speed in this region.

We next determined the effects of specific drugs on incomplete splicing. We used 5,6-dichlorobenzimidazole 1-β-D-ribofuranoside (DRB, Fig. 5c–f), (S)-(+)-camptothecin (CAM, Supplementary Fig. 7A), actinomycin D (ActD, Supplementary Fig. 7B) and 1-hydroxypyridine-2-thione zinc salt (zinc, Supplementary Fig. 7C) in commonly used concentrations. These compounds, amongst other possible routes of action, inhibit PolII elongation by inhibiting entry into elongation mode (DRB) or by blocking productive elongation (ActD)[29]. We used CAM treatment as a control to induce apoptosis[30]. Zn²⁺ cations induce hyperphosphorylation of SRSF6 and modulate its′ activity[31]. In our cell lines, both ActD and zinc resulted in a significant reduction of 5′ UTR containing sequences, comparable to the apoptosis inducing CAM treatment (Supplementary Fig. 7). Therefore, we had no confidence in interpreting these results. In contrast, 5′ UTR levels were relatively unaffected by treatment with lower concentrations of DRB (10 µM, Fig. 5c–f). Interestingly, DRB treatment increased the amount of correctly spliced exon 1-exon 2 containing sequences (Fig. 5d), which was mirrored by a reduction in the

levels of the incompletely spliced product (Fig. 5e, f), suggesting a shift towards canonical splicing.

To further strengthen our hypothesis, that incomplete splicing is influenced by PolII transcriptional speed, we introduced 'road blocks' into our minigenes using CRISPR/Cas9[32]. We used four different guide RNAs (gRNA) to tether a nuclease deficient Cas9 enzyme (dCas9) to the respective target sequences in the minigenes (Fig. 6a). The binding of CRISPR/Cas to DNA is non-covalent and transient. Rather than completely blocking transcription, we hypothesized that these tethered CRISPR/Cas complexes would hinder PolII transcription and increase the level of incomplete splicing observed in the (CAG)₁₀₀ lines (Fig. 5a). As a proof of principle, we tethered the complexes to the 5′ UTR (gRNA I, Fig. 6b), which should decrease the level of overall transcription, because PolII promoter escape/elongation is hindered at a very early stage. Indeed, we observed a reduction in minigene transcript levels for the CAG₁₀₀, but not the (CAG)₇ line (Fig. 6b, left panel 5′UTR and intron 1), possibly because PolII is already affected in the (CAG)₁₀₀ lines (Fig. 5a). The splicing pattern of the minigene transcripts was unaffected for both CAG repeat lengths (Fig. 6b, right panel, transcript levels were normalized to 5′ UTR levels). To modulate transcription further down the minigenes, we tethered the complexes with gRNAs II and III, respectively, but neither changed overall transcription, nor splicing patterns (Fig. 6c, d). On the other hand, the gRNA IV 'road block' targeted the region where we had observed statistically significant differences in the PolII occupancy between the (CAG)₁₀₀ and (CAG)₇ lines (Fig. 5a), and this led to statistically significant higher levels of incompletely spliced product (Fig. 6e).

In summary, our data indicate that slower PolII transcription rates favor the production of the incompletely spliced *HTTexon1*

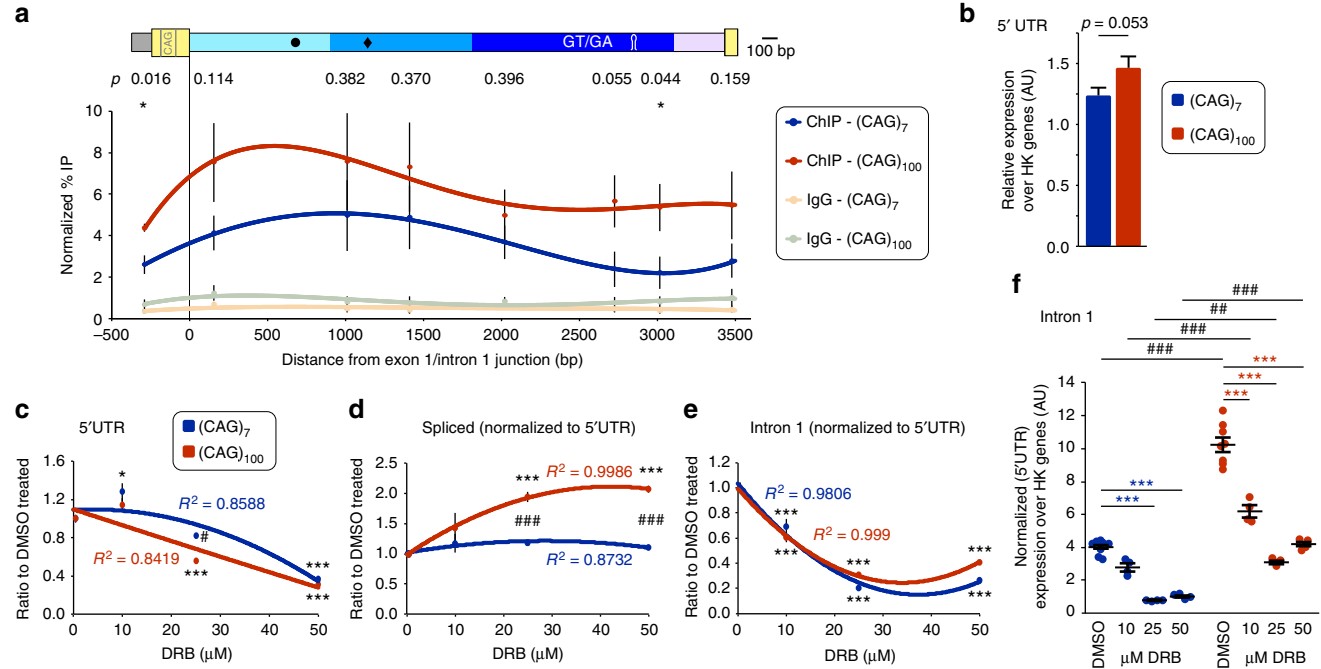

**Fig. 5** PolII transcriptional speed modulates the amount of incomplete splicing of *Htt*. **a** Chromatin immunoprecipitation of minigene associated RNA PolII. The schematic shows the position of the qPCR assays on the minigene (to scale). Data are mean ± s.e.m.; $n \geq 5$ independent ChIPs/CAG-length, Student's *t*-test. Regression fits were calculated from the individual data points. CAG-length: *$p < 0.05$. Data represents % recovered PolII normalized to input transcript levels (see also methods section). **b** 5′ UTR levels in the $(CAG)_7$ or $(CAG)_{100}$ expressing minigene lines. Data are mean ± s.e.m.; $n \geq 22$ independent experiments, Student's *t*-test. **c–f** 5,6-dichlorobenzimidazole 1-β-ᴅ-ribofuranoside (DRB) treatment. Data are mean ± s.e.m.; $n \geq 4$ independent treatments/CAG-length/concentration, two-way ANOVA with Tukey post hoc test. Regression fits were calculated from the individual data points. Treatment: *$p < 0.05$, ***$p < 0.001$. Treatment x CAG-length #$p < 0.05$, ##$p < 0.01$, ###$p < 0.001$. **c–e** Data are shown as the ratio to the respective DMSO treated sample. **c** 5′ UTR levels indicate either less transcription of the minigenes at higher DRB concentrations, cytotoxicity at higher DRB concentrations, or a combination of both. **d** Normalized spliced exon 1-exon 2 levels were significantly higher at high DRB concentrations in the $(CAG)_{100}$ versus the $(CAG)_7$ lines. **e** The reduction of intron 1 containing sequences was very similar for both CAG repeat lengths. **f** Treatment with DRB resulted in a significant reduction of minigene intron 1 containing sequences for both CAG repeat lengths

transcript. The slower transcription could provide a kinetic window for polyadenylation factors to recognize and act on the cryptic polyA sites in *Htt* intron 1.

## Discussion

We have developed a minigene system that recapitulates the incomplete splicing of *HTT*, and has allowed us to define the sequences necessary for this process. As in the case of HD mouse models and HD patient tissue, the levels of incomplete splicing increased with increasing CAG repeat length, and we showed that the spatial relation of the CAG repeat to the 5′ splice site determines the amount of incomplete splicing that occurs. Consistent with our hypothesis, the degree of incomplete splicing could be modulated by increasing or decreasing the levels of the splicing factor SRSF6. Finally, we demonstrated that the length of the CAG repeat governs the rate at which transcription occurs along the minigene. Targeting PolII with drugs and introducing obstacles along the gene influenced PolII elongation speed and, through this, modulated the level of incomplete splicing.

We generated stably integrated minigene lines, based on the mouse *Htt* sequence, and analyzed them in a human cell-background. The minigenes contained intronic canonical splicing elements and differed only in the length of the 5′ region of intron 1 (Fig. 1b). We could show that the inclusion of both cryptic polyA sites, and potential transcription attenuators (long intron construct), had the effect that the majority of the minigenes were incompletely spliced, irrespective of the length of the CAG repeat (Fig. 2). The level of incomplete splicing increased with a $(CAG)_{100}$ repeat, to the extent that the level of intron 1 sequences

was equivalent to that in the exon 1–intron 1 only constructs (Fig. 2k), in which no splicing could take place.

The general splicing factor SRSF6 binds to transcripts with an elongated CAG repeat[13]. Here, we have shown that modulation of its levels (Fig. 3) influences the extent of incomplete splicing. Overexpression greatly increased the incomplete splicing of minigenes (Fig. 3e), while knock-down resulted in reduced levels of *HTTexon1* (Fig. 3i). Could the sequestration of SRSF6 to the CAG repeats lead to splicing changes other than those of *Htt* itself? It is unlikely that this contributes to the tremendous amount of transcriptional dysregulation that develops with disease progression in HD. Incomplete splicing of *Htt* occurs at ages before the onset of transcriptional dysregulation in HD mouse models, and even at later ages there seem to be no general changes in the splicing machinery (Supplementary Fig. 5).

If SRSF6 binds to expanded CAG repeats, and results in aberrant splicing in HD, might this also be true for the other CAG-repeat diseases? To date, splicing alterations have only been identified for spinocerebellar ataxia type 3 (SCA3) in the ataxin 3 (*ATXN3*) gene. In SCA3 a 3′ truncated transcript has been found in YAC mice harboring the full length human *ATXN3* gene and in patient samples[33]. Analysis of a knock-in mouse model of SCA3 suggested that the mechanism for the generation of this transcript might involve incomplete splicing[34]. However, this SCA3 mouse model was later found to harbor a duplication of the CAG repeat and, when corrected, this mouse no longer exhibited the splicing changes[35,36]. The authors proposed that the thymidine kinase (TK) gene, which was retained between the duplicated repeat, increased the

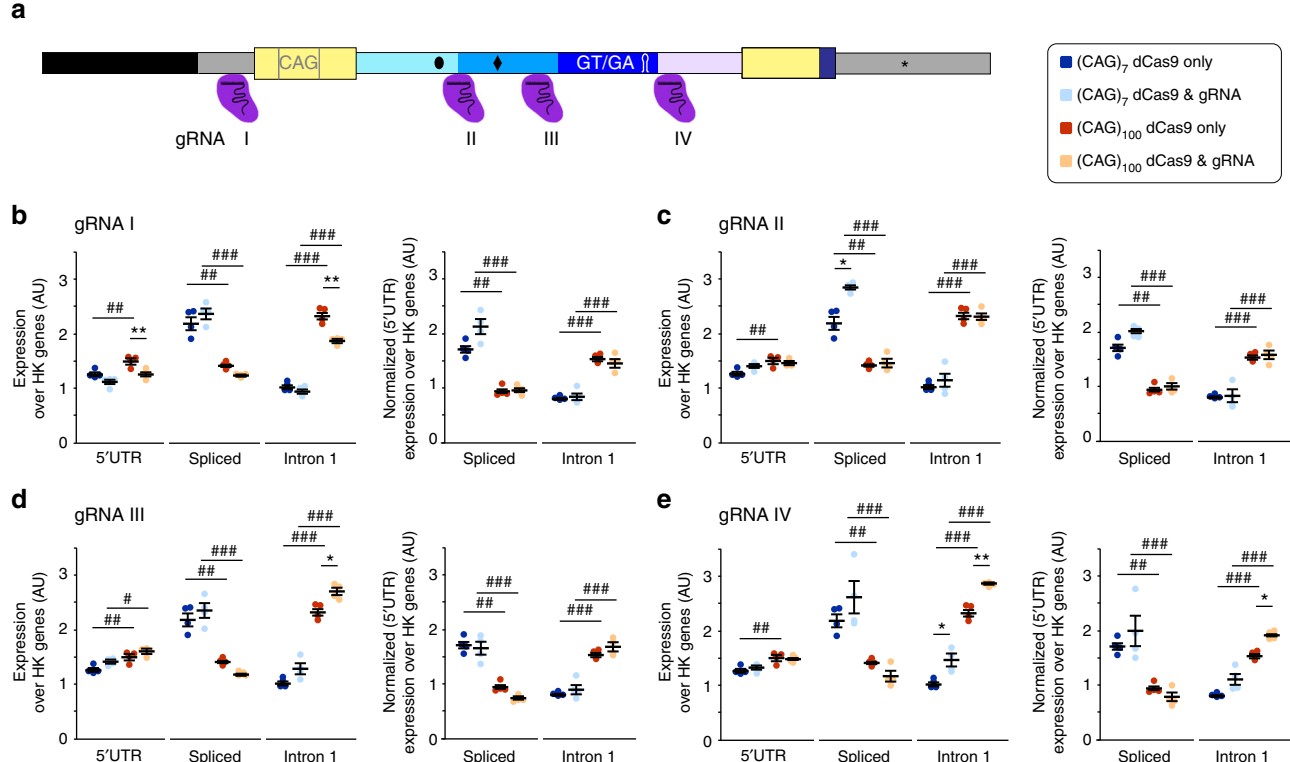

**Fig. 6** CRISPR/dCas9 induced obstacles of transcription increase the amount of incomplete splicing of *Htt*. **a** Schematic showing the binding sites of the four different guide RNAs (gRNA I-IV). **b–e** Transcript analysis of minigene sequences. Left panels show expression levels standardized to housekeeping genes. Data in the right panels are additionally normalized to 5′ UTR levels for the respective gRNA treatment. Individual data points and the mean ± s.e.m. are shown. $n = 4$ independent experiments/CAG-length/gRNA, two-way ANOVA with Tukey post hoc test. gRNA for given CAG-length: *$p < 0.05$, **$p < 0.01$, ***$p < 0.001$. gRNA x CAG-length #$p < 0.05$, ##$p < 0.01$, ###$p < 0.001$. **b** gRNA I treatment reduced the level of minigene transcripts (5′ UTR). gRNA II **c** and gRNA III **d** treatment had no significant effect on transcript levels. **e** gRNA IV treatment resulted in a significant increase of intron 1 containing sequences

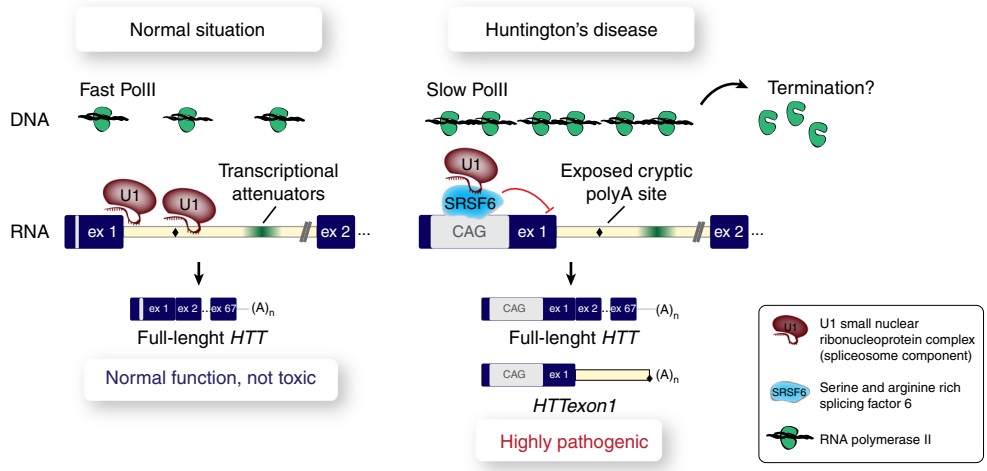

**Fig. 7** Schematic depicting mechanisms influencing incomplete splicing of *HTT* in HD. In the normal situation (left panel), U1 snRNP protects cryptic polyA sites from being utilized and defines the 5′ splice site at the exon 1/intron 1 junction. Transcription by PolII is fast and thus *HTT* is properly spliced and the full-length mRNA is produced. In HD (right panel), there is an increased amount of SRSF6 binding to the elongated CAG repeat, which could sequester U1 snRNP resulting in interference with 5′ spliceosome formation and cryptic polyA site de-protection. In addition, transcription throughout the 5′ end of the gene is slower, kinetically allowing a higher probability of cryptic polyA site usage. Together, these mechanisms lead to the generation of *HTTexon1*

amount of mis-spliced *Atxn3*[35]. Intriguingly, this might have generated a similar 'road block' for transcription of *Atxn3* as to that which we have accomplished by tethering the dCas9 enzyme to the *Htt* minigenes (Fig. 6). This 'road block' could lead to premature termination of transcription, or attenuation of elongation rates, and thus kinetically allow the generation of the truncated transcript before the intron is spliced out.

We have shown that the transcription of minigenes that contain an elongated CAG repeat is slower than that of minigenes containing short repeats (Fig. 5a, b). The current hypothesis of transcriptional termination, is that PolII pauses due to a combination of chromatin marks, DNA:RNA hybrid structures (R-loops) and termination factors[24]. Pausing of PolII is also linked to splicing. PolII transiently pauses at the 3′ end of introns[37], at terminal exons[38] and there is a polyA site dependent control step that induces premature transcription termination[28]. In the mouse *Htt* minigenes, the intronic region that includes a GT/GA microsatellite and a predicted hairpin structure was necessary to induce incomplete splicing (Fig. 1). This microsatellite does not exist in the human *HTT* gene, where the cryptic polyA site that is utilized is 7327 bp into intron 1. However, there is a CA/TA microsatellite (8443 bp from the exon 1/intron 1 junction) followed by a predicted hairpin (at 9057 bp). Both elements, microsatellites and stem-loops, have been shown to be sufficient to terminate transcription in some eukaryotic genes[39,40]. CA expansions that are not bound to the protective hnRNP L[41] lead to cleavage of pre-mRNA upstream of the expansion[42]. Taken together, these elements in intron 1 of the *HTT* gene could attenuate/terminate transcription elongation and open up a kinetic window for utilization of the cryptic polyA sites.

Negative elongation factor (NELF) and DSIF (5,6-dichloro-1-β-D-ribofuranosylbenzimidazole (DRB) sensitivity inducing factor) are both needed for PolII pausing at promoter proximal sites[43]. DSIF comprises the mammalian homologs of SPT4 and SPT5[44]. DRB (Fig. 5) inhibits the elongation factor P-TEFb through inhibition of cyclin-dependent kinase 9 (CDK9), which in turn causes an increase in the DSIF mediated pausing of PolII[45,46]. Such promoter proximal pausing at the *HTT* locus could be reflected by the increased amount of PolII observed at 5′ UTR sequences (Fig. 5a). RNA PolII holoenzyme needs to acquire the specific elongation factor SPT4/SUPT4H1 (SPT4 homolog, DSIF elongation factor subunit) to be able to transcribe efficiently through CAG repeats[47]. Increased interaction of DSIF, in particular the SPT4 subunit, with PolII by DRB treatment could improve transcription through the repeat, and consequently, throughout the gene. This would decrease the likelihood of cryptic polyA site usage and decrease the amount of incomplete splicing (Fig. 5d–f). Furthermore, DRB blocks the phosphorylation of DISF (SPT5 subunit) through inhibition of CDK9, which is a prerequisite for PolII to pass the polyA checkpoint of transcription termination[28]. Therefore, DRB treatment could lead to PolII not passing the polyA checkpoint at the cryptic polyA site in intron 1 and thus effectively blocking the maturation of *HTTexon1*, shifting the steady state levels towards properly spliced *Htt* (Fig. 5d–f).

In summary, using our model system of incomplete splicing of *Htt*, we uncovered multiple regulatory mechanisms that influence the amount of *HTTexon1* production (Fig. 7). We showed that a precise spatial relation of the CAG repeat to the 5′ splice site and specific intronic sequences was necessary for the formation of *HTTexon1*. Furthermore, modulation of the levels of SRSF6, a general splicing factor, regulated the levels of *HTTexon1*. Finally, we demonstrated that PolII transcription elongation and termination determine if the cryptic polyA sites in intron 1 are utilized and *HTTexon1* is produced. These findings unravel some of the fundamental requirements for incomplete splicing of *HTT* leading to a more detailed view of the processes that drive HD pathogenesis. It has been shown that stabilizing the 5′ splice site spliceosome assembly with a small molecule at the exon 7-intron 7 junction of the SMN2 gene (survival of motor neuron 2) resulted in a marked increase of incorporation of exon 7 into the mature mRNA[48,49]. Our system could provide the basis of a high-

throughput screen for small molecules that modulate incomplete splicing and thereby help to develop future therapeutics for HD.

## Methods

**Generation and maintenance of stable cell lines**. We used the Flp-In™ system (ThermoFisher) to generate all cell lines. This system permits the generation of stable cell lines within an isogenic background that only differs due to the integrated sequence. We used the Flp-In™ −293 human embryonic kidney cell line as the parent line (R750-07, ThermoFisher). All cell culture supplies were purchased from ThermoFisher. The integration procedure of the cell lines was performed according to the manufacturer's instructions. The stably integrated cell lines were selected with 100 μg/ml hygromycin B (10687010). We expanded at least 3 single cell clones through four passages to a T75 stage, which was designated passage 0. Every cell line was repeat sized (as described in[14]) at this stage to ensure that lines had the desired CAG repeats and to exclude those that had changed through CAG repeat instability during the integration procedure. Growth medium was DMEM (11960085), 10% (v/v) FBS (16000044), 2 mM L-glutamine (25030024), 100 U Pen/Strep (15140122), 100 μg/ml hygromycin B (10687010). Cells were passaged when they reached approximately 90–95% confluence, usually every 3 days. All cell lines were routinely checked for mycoplasma contamination.

**Generation of plasmids used in this study**. All *Htt* sequences are based on NCBI accession number 15194 (murine *Htt*), genome build NC_000071.6 bases 34760737 to 34913521. We used the Herculase (Agilent) PCR system for cloning according to the manufacturer's protocol. In cases where CAG repeats longer than 40 CAGs were amplified, 5% dimethyl sulfoxide was added to the PCR reactions.

The following plasmids were generated by cloning PCR products from wild type genomic DNA (CBA × C57BL/6) into pCR2.1-TOPO-TA vectors (Life Technologies). Primers are given in brackets (see also Supplementary Table 1): about 1 kb of murine *Htt* promoter (pHtt1_f, pHtt1_r) = pHtt1; 5′ UTR, exon1 and the first 98 bp of *Htt* intron 1 (pHtt2_f, pHtt2_r) = pHtt2; the following 819 bp of *Htt* intron 1 after the end of pHtt2 (pHtt3_f, pHtt3_r) = pHtt3; the following 3054 bp of *Htt* intron 1 after the end of pHtt2 (pHtt3_f, pHtt30_r) = pHtt30; the last 338 bp of *Htt* intron1, exon2, FLAG-tag (pHtt6_f, pHtt6_r) = pHtt6; about 650 bp of the 3′UTR of exon 67 of *Htt* (pHtt7_f2, pHtt7_r2) including the endogenous polyA site = pHtt7. All plasmid sequences were validated by sequencing: pHtt1: from the NotI site C426G and G764T; pHtt2: no change; pHtt3: from the KpnI site T300A and T461C; pHtt30: from the KpnI site T300A, T461C, T2697G, T2843C; pHtt6: no change; pHtt7: no change. Some nucleotide changes like T300A, T461C from the KpnI site were found in independently amplified and cloned plasmids. They therefore most likely represent deviations from the NCBI deposited mouse genome due to the different mouse strains and not PCR introduced mutations.

To generate the backbones for the integration plasmids with different intron lengths, the CMV-promoter was cut from pcDNA5/FRT (Life Technologies) with SpeI/EcoRV, the plasmid was re-ligated and sequenced to give pHtt8. pHtt1 was subcloned with NotI/PmeI into pHtt8 to give pHtt9. pHtt2 was subcloned with BamHI/PmeI into pHtt9 to give pHtt10. pHtt6 was subcloned with MfeI/BamHI into pHtt7 to give pHtt11. pHtt3 was subcloned with KpnI/MluI into pHtt11 to give pHtt12. pHtt12 was subcloned with KpnI/PmeI into pHtt10 to give pHtt15 (short intron backbone, contains (CAG)₇ repeat). A PCR from pHtt11 (pHtt32_f, pHtt7_r2) was cloned into pCR2.1-TOPO-TA to give pHtt32. pHtt30 was subcloned KpnI/SbfI into pHtt32 to give pHtt14. pHtt14 was subcloned KpnI/PmeI into pHtt10 to give pHtt17 (long intron backbone, contains (CAG)₇ repeat). pHtt30 was cut with SacI/SbfI, re-ligated and sequenced to give pHtt13. pHtt13 was subcloned KpnI/PmeI into pHtt10 to give pHtt16 (medium intron backbone, contains (CAG)₇ repeat).

To generate different CAG repeat lengths, PCR products consisting of 5′ UTR, exon1 and the first 98 bp of *Htt* intron 1 (pHtt2_f, pHtt2_r) from genomic DNA of different knock-in lines were cloned into pCR2.1-TOPO-TA. Template: HdhQ50 = pHtt18; HdhQ100 = pHtt19; HdhQ150 = pHtt20. CAG repeat lengths in the range from 20 to 45 were obtained by random priming in the CAG repeat from an HdhQ50 genomic DNA template in the first PCR (two separate PCRs: pHtt2f/CTG7 and CAG7/pHtt2r). The PCR products were purified and used as templates in a second PCR (pHtt2f/pHtt2r). Products of different sizes were cloned into pCR2.1-TOPO-TA. All plasmid sequences were validated by sequencing and showed no changes. Each BamHI/KpnI fragment was subcloned into the backbones of pHtt15 (short), pHtt16 (medium) and pHtt17 (long) to give the final integration plasmids. pHtt21 = (CAG)₅₀ short intron; pHtt22 = (CAG)₁₀₀ short intron; pHtt23 = (CAG)₁₅₀ short intron; pHtt24 = (CAG)₅₀ medium intron; pHtt25 = (CAG)₁₀₀ medium intron; pHtt26 = (CAG)₁₅₀ medium intron; pHtt27 = (CAG)₅₀ long intron; pHtt28 = (CAG)₁₀₀ long intron; pHtt29 = (CAG)₁₅₀ long intron; pHtt37 = (CAG)₃₂ long intron; pHtt38 = (CAG)₂₃ long intron; pHtt39 = (CAG)₄₁ long intron; pHtt40 = (CAG)₃₇ long intron; pHtt41 = (CAG)₂₃ medium intron; pHtt42 = (CAG)₄₁ medium intron.

To generate plasmids that only expressed exon 1 sequences, the parent vectors with different CAG repeat lengths were cut with NheI/PmeI and re-ligated. pHtt33 = (CAG)₇; pHtt34 = (CAG)₅₀; pHtt35 = (CAG)₁₀₀; pHtt36 = (CAG)₁₅₀.

To generate integration plasmids to study the position dependency of the CAG repeat, PCR products from a (CAG)$_{23}$ and a (CAG)$_{100}$ containing plasmid were cloned into pCR2.1-TOPO-TA and sequenced. In addition to the CAG repeat, the plasmids contained 19 bp adjacent upstream and 18 bp adjacent downstream sequences. Primers for BamHI integrated CAG repeats were BamHIF/BamHIR: KpnI: KpnIF/KpnIR; HincII: HincIIF/HincIIR; NdeI: NdeIF/NdeIR. All plasmid sequences were validated by sequencing and showed no changes.

Human *SDHA* (NCBI gene ID: 6389) and *SPP1* (NCBI gene ID: 6696) minigenes were cloned from genomic DNA extracted from the HEK293 cell line. Both PCR products (SDHA: SDHAf/SDHAr; SPP1: SPP1f/SPP1r) were first cloned into pCR2.1-TOPO-TA and the 5′ and 3′ ends of the minigenes were confirmed by sequencing. The CAG repeats were introduced and subsequently, the constructs were subcloned KpnI/BamHI into pCR5/FRT. The minigenes were under the control of a CMV promoter and a bGH terminator. Furthermore, in all constructs an ATG start codon was introduced 5′ of the first cloned exon.

Mouse (NCBI gene ID: 67996) and human (NCBI gene ID: 6431) *SRSF6* overexpression plasmids were cloned including a 5′ Kozak sequence. Mouse or human cDNA, respectively, was used as the PCR template. Primers were: human *SRSF6*: hmSF6f-K/hSF6r; mouse *Srsf6*: hmSF6f-K/mSF6r. The PCR products were cloned BamHI/NotI into pcDNA3. Constructs were under the control of a CMV promoter.

pAC84-pCR8-dCas9 was a gift from Rudolf Jaenisch (Addgene plasmid # 48218)[50]. MLM3636 was a gift from Keith Joung (Addgene plasmid # 43860). The coding sequence for dCas9 was transferred to the Gateway™ pT-Rex™-DEST30 Vector (ThermoFisher) according to the manufacturer's instructions. MLM3636 was digested with BsmBI and the four gRNA were introduced. To this end, complementary oligonucleotides (see Supplementary Table 1) were annealed and ligated into the cut MLM3636. The expression was driven by the human U6 promoter. gRNA plasmids were sequenced and contained no mutations.

Overexpression plasmids for SRSF1, SRSF2 and SRSF3 were obtained from Addgene. SRSF1 was a gift from Honglin Chen (Addgene plasmid # 99021)[51]. SRSF2 was a gift from Kathleen Scotto (Addgene plasmid # 44721)[52]. SRSF3 was a gift from David Bartel (Addgene plasmid # 46736)[53].

**Transfection, siRNA, CRISPR/dCas9 and drug treatments**. $2.5 \times 10^5$ cells were seeded in a 6-well plate. Experiments were initiated on the following day, and cells were analyzed 48 h later for transfections, or 6 h later for drug treatments. All DNA/RNA transfections were performed with the jetPRIME® system (Polyplus-transfection) according to the manufacturer's instructions. For plasmid transfections, 1 µg of plasmid was transfected. For siRNA transfections, 30 nM final concentration siRNAs were transfected. For CRISPR/dCas9 experiments, 1 µg dCas9 plasmid and 1 µg of each gRNA plasmid were transfected. Drugs were solubilized in 100% dimethyl sulfoxide (DMSO) as stock solutions, aliquoted and kept at −20 °C. Stocks solutions were: 5,6-dichlorobenzimidazole 1-β-D-ribofuranoside (DRB) 10 mM; actinomycin D (ActD) 1 mM; (S)-(+)-camptothecin (CAM) 10 mM; 1-hydroxypyridine-2-thione zinc salt (zinc) 50 mM. Working solutions were prepared on the day of the experiment and pre-diluted with 100% DMSO to a 200-fold higher concentration than the desired final concentration. Finally, 10 µl of the drug solutions, or 100% DMSO (0.5% final concentration of DMSO) were added to a 6 well plate (2 ml medium), mixed and incubated for 6 h at 37 °C.

**Quantitative real-time PCR and 3′RACE**. RNA was extracted using QIAZOL together with RNeasy Mini kits (Qiagen) according to the manufacturer's instructions. 2–4 µg total RNA was reverse transcribed with MMLV (Invitrogen, Moloney murine leukemia virus) using the UAPdT primer and according to the manufacturer's instructions. For 3′RACE analysis, the RT reaction mix was digested with 1U of RNase H (Invitrogen) for 1 h at 37 °C. The cDNA was subsequently diluted 1:10 in water and 2 µl were used as template. For qPCR analysis, the RT reactions were diluted with water (1:10) and analyzed on a CFX96 (Bio-Rad). For details about primers and probes see Supplementary Table 2. We used Taqman RT quantitative PCR (qPCR) with multiplexed assays. Multiplexing was as follows (fluorophore in brackets): *ACTB* (FAM) + splice ex1/ex2 −19f (TexasRed) + 5′ UTR (Cy5.5); *ATP5B* (FAM + *HTT* ex66/67 (Cy5.5); *SDHA* (FAM) + intron 1 (TexasRed) + spliced ex1/ex2 −34f (Cy5.5). All assays were tested for reproducibility of the results in the multiplexed versus single runs. All other assays were performed as single runs. For Fig. 1d–f, the geometric mean of the two spliced ex1/ex2 (−19f and −34f) assays was used to generate the graphs. For all other figures only the spliced ex1/ex2 −19f was used. Evaluation of the data was performed using the ΔΔCt evaluation method[54]. The geometric mean of the expression levels of the following housekeeping genes was used to standardize the samples: Figs. 1d–f, 3a, b, 3d, e, 4b, c, 5b, 6b–e, Supplementary Fig. 3, Supplementary Fig. 4, Supplementary Fig. 6A-6D: *ATP5B*, *ACTB* and *SDHA*, Fig. 6c–f and Supplementary Fig. 6C-6E: *ATP5B* and *ACTB*. Primers used for 3′RACE and qPCR are listed in Supplementary Table 1.

**Prediction of polyA sites and terminators and sequencing**. To predict polyA sites in intron 1 of *Htt*, we used the SoftBerry polyAH algorithm (http://linux1.softberry.com/all.htm). Transcription terminator like regions were predicted using ARNold[55]. Sequencing was performed on a ABI3730xl DNA analyzer using the Big Dye Terminator 3.1 mix (ABI) according to the manufacturer's protocol.

**3B5H10 immunoprecipitation, western blotting and antibodies**. HTT immunoprecipitation (IP) was carried out with 3B5H10 (Sigma-Aldrich) coupled magnetic beads (M-270 Epoxy, ThermoFisher). The coupling procedure was as described in[13]. Cells were lysed in ice-cold NET buffer (50 mM HEPES pH 7.4, 100 mM NaCl, 0.5% (w/v) Triton X-100, 1 mM EDTA) supplemented with cOmplete™ protease inhibitors (Sigma-Aldrich). 25 µg of total lysate was collected as inputs. 500 µg total protein extract was mixed with an equal volume of dilution buffer (50 mM HEPES pH 7.4, 220 mM NaCl, 1% (w/v) Triton X-100, 20 mM EDTA, 0.2% (w/v) SDS, 0.4% (w/v) Na-deoxycholate, 4 mM DTT, 1:500 PMSF supplemented with cOmplete™ protease inhibitors (Sigma-Aldrich)). 10 µl 3B5H10 coupled magnetic beads were added and the mix was incubated for 4 h at 4 °C with slight agitation. IPs were washed 4 times with each 0.4 ml of wash buffer (50 mM HEPES pH 7.4, 160 mM NaCl, 1% (w/v) Triton X-100, 10 mM EDTA, 0.1% (w/v) SDS, 0.2% (w/v) Na-deoxycholate, 2 mM DTT, 1:1000 PMSF supplemented with cOmplete™ protease inhibitors (Sigma-Aldrich)). Bound proteins were eluted by heating the washed beads with HU buffer (200 mM Tris-Cl pH 6.8, 8 M urea, 5% (w/v) SDS, 1 mM EDTA pH 8.0, 215 mM β-mercaptoethanol) for 10 min at 65 °C. Inputs were also mixed with an equal amount of HU buffer.

Samples from the immunoprecipitation were loaded onto a 16% poly-acrylamide gel. Samples for analysis of SR protein expression were loaded onto a 12% poly-acrylamide gel. Separated proteins were transferred onto a nitrocellulose membrane (Bio-Rad) and blocked for 1 h with blocking buffer (5% (w/v) skimmed milk powder in TBS-T (50 mM Tris-Cl pH 7.4, 150 mM NaCl, 0.1% (w/v) Tween 20)). All blots were incubated with primary antibodies over night at 4 °C in TBS-T. Wash buffer was TBS-T. Secondary antibodies (1:5000 in TBS-T for 45 min at room temperature) were purchased from LI-COR and western blots were visualized on an Odyssey Sa (LI-COR) and analyzed with the Image Studio Lite Ver 3.1 (LI-COR). Antibody dilutions were as follows: anti-FLAG (2368P, Cell Signaling) 1:100; anti-HTT (MW8) (CHDI-90000942-1, CHDI Foundation) 1:250; anti-HTT (S830) (in-house) 1:500; anti-SRSF6 (ab140623, Abcam) 1:1000; anti-SRSF1 (sc-33652, Santa Cruz Biotechnology) 1:500; anti-p-SRSF2 (sc-53518, Santa Cruz Biotechnology) 1:500; anti-SRSF3 (sc-398541, Santa Cruz Biotechnology) 1:500; anti-SDHA (ab14715, Abcam) 1:10000.

**Chromatin immunoprecipitation**. $1 \cdot 10^6$ cells were seeded in a 150 mm petri dish with 40 ml of growth medium and 80–90% confluence was reached after 3 days of incubation. DNA and bound proteins were cross-linked with 1% formaldehyde (SigmaAldrich) for 5 min at 30 °C on an orbital shaker (150 rpm). The crosslinking reaction was quenched by addition of 125 mM glycine for 5 mins at RT on an orbital shaker (150 rpm). The cells were rinsed with DPBS (14190250, Thermo-Fisher), scraped off, transferred into a test tube and pelleted (RT, 2 min, 500 g). The pellet was lysed with 200 µl lysis buffer (50 mM Tris-Cl pH 8.0, 1% (w/v) SDS, 10 mM EDTA pH 8.0, supplemented with cOmplete™ protease inhibitors (Sigma-Aldrich)) and incubated for 5 min on ice. The cross-linked DNA was sheared by sonication in a bioruptor sonicator (UCD-200 TO, Diagenode) with settings 30 s on and 60 s off, high intensity, 20 cycles. The resulting fragment length was checked on 2.5% agarose gel and was about 300 bp. The sonicated lysate was centrifuged for 7 min at 4 °C. The DNA concentration was measured against the SDS lysis buffer on a nanodrop 1000 (ThermoFisher). Immunoprecipitation (IP) was carried out with 100 µg of chromatin. 2 µg of each antibody, anti-RNA polymerase II CTD repeat YSPTSPS (phospho S2) (ab5095, Abcam) and rabbit IgG (2729, Cell Signaling) were used for IP. The samples were mixed with 4 times the volume of ChIP dilution buffer (20 mM Tris-Cl pH 8.0, 0.1% (w/v) SDS, 1% (w/v) Trition X-100, 150 mM NaCl, 2 mM EDTA pH 8.0, supplemented with cOmplete™ protease inhibitors (Sigma-Aldrich)) and incubated overnight at 4 °C on a rotating wheel. Prior to purifying the antibody bound complexes, 5% input was taken from all samples for data normalization. Pre-washed (SDS lysis buffer) dyna protein G beads were then added to IPs and incubated for 3 h at 4 °C on turning wheel. The IP reactions were washed for 5 min at RT with 0.5 ml of each wash buffer: 2 times low salt (50 mM HEPES pH 7.4, 150 mM NaCl, 0.1% (w/v) sodium deoxycholate, 1% (w/v) Triton X-100, 1 mM EDTA pH 8.0), high salt (50 mM HEPES pH 7.4, 500 mM NaCl, 0.1% (w/v) sodium deoxycholate, 1% (w/v) Triton X-100, 1 mM EDTA pH 8.0), LiCl (10 mM Tris-Cl pH 8.0, 250 mM LiCl, 0.5% (w/v) sodium deoxycholate, 0.5% (w/v) NP-40, 1 mM EDTA pH 8.0)) and 2 times TE (10 mM Tris-Cl pH 8.0, 1 mM EDTA pH 8.0). IPs were eluted twice with SDS elution buffer (50 mM Tris-Cl pH 8.0, 1% (w/v) SDS, 10 mM EDTA) for 20 min at 70 °C. The combined elutions were incubated overnight at 70 °C. The next day, 1.5 µg of RNase A (ThermoFisher) per IP was added and incubated at 37 °C for 30 min. 40 µg glycogen (ThermoFisher) and 60 µg proteinase K (ThermoFisher) were added and the reactions were incubated at 37 °C for 2 h. Purified DNA was extracted using the Qiagen PCR purification kit (28104) and resuspended in 10 mM Tris pH 8.0 for further analysis.

**Analysis of transcriptomic data bioinformatics analysis**. Count data for the zQ175 mouse datasets were obtained from the HDinHD website (www.hdinhd.org, downloaded on 17/01/2018)[56]. For the human *post-mortem* cortex data, sra files (GSE79666) were downloaded from GEO (www.ncbi.nlm.nih.gov/geo), mapped to the GRCh38 release 90 human transcriptome (www.ensembl.org) with STAR v2.5.3a[57] and quantified with Salmon v0.8.2[58]. All datasets were analyzed for batch effects, gender corrected (limma v3.34.8[59]) and outliers removed. Dysregulation

was computed using tximport v1.6.0[60] and DESeq2 v1.18.1[61]. Spliceosome components were downloaded from KEGG (www.genome.jp/kegg; mmu03040; hsa03040) and the transcript dysregulation data was mapped onto the common genes. Networks were generated with Cytoscape v3.6.0[62].

**Statistical analysis**. All data were screened for outliers using a Grubbs' test (GraphPad). All data were analyzed with IBM SPSS using a two-tailed homoscedastic Student's *t*-test or a two-way ANOVA with either Tukey or Bonferroni post hoc test as specified in the figure legends. *P*-values < 0.05 were considered to be statistically significant. To determine the type of regression fit for data in Figs. 2, 5 and in Supplementary Fig. 3, we calculated regression fit statistics and chose the most significant fit. Regression fits were calculated from the individual data points. Regression statistics can be found in Supplementary Table 3.

## Data availability

Count data for the zQ175 mouse datasets were obtained from the HDinHD website (www.hdinhd.org, downloaded on 17/01/2018). The human post-mortem cortex data were downloaded from GEO (www.ncbi.nlm.nih.gov/geo, GSE79666). The authors declare that the data supporting the findings of this study are available within the paper and its supplementary information files. Scripts for the bioinformatics analysis are freely available upon request from the corresponding authors.

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

## Acknowledgements

We would like to thank Prof. Nicholas Proudfoot and Dr. Natalia Gromak for helpful discussions and advice. The authors would also like to thank Ms. Sabine Huber for help in establishing the drug treatments of the cell lines. We thank the Jaenisch, Chen, Scotto, Bartel and Joung labs for the donation of plasmids. The authors acknowledge support by the High Performance and Cloud Computing Group at the Zentrum für Datenverarbeitung of the University of Tübingen (BwForCluster BinAC), which is funded by the state of Baden-Württemberg through bwHPC and the German Research Foundation (DFG) through grant no INST 37/935-1 FUGG. This work was supported by funding from the MRC (MR/L003627/1) and the CHDI Foundation.

## Author contributions

Conceptualization, A.N. and G.P.B.; Methodology, A.N.; Formal Analysis, A.N.; Investigation, A.N. and A.A.D.; Supervision, A.N. and G.P.B; Resources, A.C.B.; Visualization, A.N.; Writing—Original draft, A.N. and G.P.B.; Writing—review & editing, A.N., A.A.D., A.C.B. and G.P.B.; Funding acquisition A.N. and G.P.B.

## Additional information

**Competing interests:** The authors declare no competing interests.

