## [Peer Review File · Nature Communications]

Reviewers' comments:

Reviewer #1 (Remarks to the Author):

Huntington's disease (HD) is caused by a CAG repeat expansion in HTT exon 1 (E1) and previous studies have shown that the expansion induces a block to E1-E2 splicing with the degree of blockage proportional to CAG repeat length. This altered splicing results in cryptic polyadenylation site selection in intron 1 and a truncated HTT exon1 that produces a very pathogenic protein. Here, Neueder et al use minigene reporter 293 isogenic lines to investigate this splicing block and provide evidence for their model that SRSF6 binds to CAG repeats and draws U1 snRNP away from the exon 1 5'ss. They also show that RNA pol II transcription rates influence altered splicing of Htt and that slower RNA pol II transcription, induced by targeted gRNAs and dCas9, favor incomplete splicing and HTT exon1 production. The manuscript is well written and the results are generally convincing, but I have noted several issues below that should be addressed.

Major points:

1. Fig. 2. To confirm the observed effects of SRSF6 overexpression (OE) are specific to this splicing factor, the OE effect of at least one additional SR protein should be tested.
2. Fig. 2D. The higher level of CAG100 incomplete splicing with human vs mouse SRSF6 is attributed to only partial spliceosome integration by the mouse protein. This is a reasonable conclusion since the mouse Srsf6 transcript levels were higher than human but there is no evidence showing that the corresponding mouse and human SRSF6 protein levels are similar.
3. Fig. 2E. The surprising finding that human SRSF6 levels were lower in CAG100 vs CAG7 controls (siRNAscr) should be discussed in more detail.
4. Fig. S4. The SDHA and SPP1 minigenes are used to examine CAG-induced mis-splicing but the repeats are only inserted into introns. What is the result if these CAG repeats are inserted into an exon at the same distance from the 5'ss as HTT E1?
5. Discussion. The authors' HD model illustrates SRSF6 binding to the CAG repeat causes misguiding of U1 snRNP to the expansion region versus the E1 5'ss but what is the experimental evidence that supports this CAG-SRSF6-U1snRNP complex formation?

Minor points:

6. Results P5. The original repeat series was 7, 23, 32, 41, 54 and 103 followed by generation of isogenic 293 lines that I assume resulted in clonal variation in repeat sizes (e.g., 53, 55 in Fig. S1) so for clarity it would be helpful to include a sentence to that effect.
7. Fig. S1B. The Q103 fragment appears to be the broader band in the 103 lane but this separation is unconvincing and would be clearer if the electrophoresis run time was increased.
8. Fig. 2E and P6. The text on P6 indicates that siRNA KDs of both mouse Srsf6 and human SRSF6 resulted in decreased transcript levels but only human is shown in Fig. 2E.

Reviewer #2 (Remarks to the Author):

This is an important and systematic study providing details of the essential regions of the 5'HTT gene necessary for normal splicing as well as for incomplete splicing, which was first described by this group. This incomplete splicing to create an exon1 protein is a critical new finding in the field that has significant therapeutic implications, including for HTT lowering approaches and alternative targets. Defining the critical regions extends an understanding of how this exon 1 splice variant is produced and the studies provide important insights into the mechanisms involved that can be used to develop models to ascertain whether the production of exon1 protein through alternative splicing is a required step in pathogenesis. Using the mouse mini-gene is a creative approach to investigating mechanisms and the generation of isogenic cell lines expressing these constructs is a critical step to enable quantitative analysis. The authors also test whether overexpression or

decreased SRSF6 modulates incomplete splicing of HTT and of note, there is a selective effect on incomplete splicing (versus canonical splicing) in the context of a highly expanded repeat. It is also intriguing that the position of the CAG repeats modulates the incomplete splicing phenotype. The authors then go on to show that the transcriptional speed of PolII modulates incomplete splicing and evaluate a small molecules that inhibits PolII elongation, as well as introducing adding in tethered CRISPR/Cas complexes that hinder PolII and increased incomplete splicing depending where it is introduced.

1) References missing on p2- 2nd paragraph of intro

2) It would be helpful to have a bit more description of the cell model and rationale for how developed. For instance, although it is clear that the mouse Htt minigenes were used to be able to distinguish between the minigene and endogenous HTT transcripts, it would be helpful to know how conserved the sequences in the constructs are to human HTT, given that the cell line used is human.

3) The initial description of the constructs is a bit confusing, although it becomes more clear in the results as the experiments are described. For general reader, the initial descriptions of the constructs and rationale for each needs to be a bit better clarified. For instance, the relevance of the long intron is not initially clear.

4) Does the presence of the expanded repeat in the Q100 isogenic line (or other expanded CAG repeat lines) show aggregation or other phenotypes that could be assessed in these assays as well? In particular does treatment with DRB and reduction of incomplete splicing influence aggregation.

Reviewer #3 (Remarks to the Author):

This group previously reported that alternative splicing of Htt gene produces Exon1 peptide (Htt-Exon1) without cleavage by proteases from full-length Htt protein. The discovery revived the significance of previously generated and world-widely used mouse model R6/2. In this new work, the authors investigated the transcriptional speed and skipping exon1-intron1 junction. They also showed the effect of a splicing-related factor SRSF6, which they previously reported as a factor binding to the elongated CAG repeat and affect the alternative splicing, on the skipping of the exon-intron junction. Some of their experiments included beautiful results, while it is still difficult for this reviewer to be convinced about the mechanism. There are two major points that reduces the significance of this paper.

Major points

1) Some researchers investigated the relationship between transcription speed and alternative splicing. Basically, when the speed is slowed down, the stochastic chance for splicing related factors bind to an exon-intron junction of pre-mRNA is generally increased to up-regulate the splicing efficiency between neighboring exons (Fong et al, Gen Dev 2014; Bentley, Nat Rev Genet 2014). This is a contrastive scheme against the authors' findings in this paper. Naturally there would be an exceptional case that does not follow the general rule. However, the authors should reveal more detailed mechanism, not just showing the dose relationship between SRFF6 and splicing, with more intensive data on the change in splicing machinery under CAG repeat expansion.

2) In this sense, it is not clear how SRSF6 modulates the splicing at the exon-intron junction. The slow transcription would increase the chance of SRSF6 to bind to CAG repeat sequence, but how SRSF6 inhibits the splicing. With direct evidences such as changed components of splicing complex at the Htt exon-intron junction or KD experiments in vitro, the authors should reveal the mechanism how it is related to the skipping.

Minor points

1) Some of the experiments are difficult to catch what they actually did, and detailed methods should be described. Especially, Fig 1D-F are hard to understand what methods and parameters

they actually used for calculation of the ratios and how they evaluate the significance of differences in statistics after reading Fig legend and methods.

Reviewer #4 (Remarks to the Author):

The thesis of this study is that transcriptional events affect the translation of an exon1-intron1 mutant huntingtin fragment. The importance of the study is that the mutant fragment could have important consequences on the pathogenesis of HD. This idea is interesting and important. Sections of the manuscript are devoted to the concept that the CAG repeat slows Pol II activity, thereby facilitating the production of the mutant HTT fragment from the exon1-intron1. Experiments are presented that test this concept. Clearly, the studies are involved, requiring invention of cell lines. Over-riding concerns are that the novel approaches need to be confirmed (CRISPR), that small differences (though significant) might not be biologically important, and that changes in mRNA do not necessarily correlate with changes in specific proteins (Westerns or similar). Splicing is complex, so that attribution of the changes to SRSF6 (not part of spliceosome) may be incomplete. Specific points on each Figure are meant to clarify a reader's analysis of data. Reasons to do some of the experiments should be made obvious.

Figure 1. The authors made cell lines that express long, medium, short and exon 1 only introns with associated CAG lengths in series. It is supposed that the cell lines, though easier to use and more accessible to experimentation than live tissue, will mimic the natural state of mutant HTT. Shortcomings of the results include (1) lack of detail in figure legend for D-F. In Fig 1F, is the curve (not clear how it is fit) related to changes in SFSR6, as shown in Figure 2. Northern blots would more clear show the splice isoform than would the PCR products. Figure 1D requires more detail. What is the point of the figure?

page 5, bottom. The sentence about incompletely spliced transcripts and endonucleolytically cleaved pre-mRNA is not experimentally demonstrated.

page 6 at top. Supplementary Figure 1 should be a figure in the main paper.

page 6. The last part of the last paragraph is confusing. Why is the SRSF6 lower in the CAG100 cells? This finding is potentially important to interpret the data. Are there other changes in the cells that might account for different results between the CAG 100 and CAG7 cells? In all, this paragraph is confusing. It is unclear if SFSR6 is in the spliceosome, so that the term "not fully integrate into the human spliceosome" might be misleading.

Figure 2. Compare this figure to Figure 1F. There could be variability in the SFSR6 abundance in the cells. It would be important (critical) to measure the amount of SFSR6 protein. mRNA does not always translate to protein. In knockdown studies, the Western would help to interpret the extent of change of the SFSR6 protein. Furthermore, splicing factors can be autoregulated, so that Westerns become imperative. There is no binding data shown, so that the last paragraph on page 7 is conjecture. Any splicing data should be confirmed by Northern blot, better suited to demonstrate splicing. In 2E, the difference between mRNA CAG7 and CAG100 should be confirmed by Western blot. In general, there are small differences in many of these plots -- not clear if the small differences are biologically important.

Page 8 and Figure 3. This study tested the idea that placement of CAG repeats could affect splicing. A compelling reason for this study is not presented. What is the real impact of the experiment? The reader ought to know why the experiment has a bearing on the HD pathogenesis.

Figure 4 studies the effect of the rate of transcription of Pol II on the amount of incomplete splicing. A good idea. However, the data show a series of steady state measurements, rather than

a real rate. The interpretation will be indirect. In Figure 4A, the curve fit should be clarified. The use of "regression fits" should be clarified.

Figure 5. The CRISPR/Cas9 studies test the idea that slowing transcription presents "obstacles" to increase the incomplete splicing. The figures actually show little change (although significant) and raises the concern that the changes might not be biologically important. Would a 10% change in the mRNA really change production of a mRNA, or protein, in a meaningful way? The use of CRISPR in this way would need to be better characterized as presenting an obstacle. Indeed, on page 11 it is said the "we hypothesized that these tethered CRISPR/Cas complexes would hinder Pol II transcription..". The hypothesis would need to be secured for the experiment to be interpretable.

The discussion is difficult to interpret based on the data as presented.

Reviewers' comments:

Reviewer #1 (Remarks to the Author):

Huntington's disease (HD) is caused by a CAG repeat expansion in HTT exon 1 (E1) and previous studies have shown that the expansion induces a block to E1-E2 splicing with the degree of blockage proportional to CAG repeat length. This altered splicing results in cryptic polyadenylation site selection in intron 1 and a truncated HTT exon 1 that produces a very pathogenic protein. Here, Neueder et al use minigene reporter 293 isogenic lines to investigate this splicing block and provide evidence for their model that SRSF6 binds to CAG repeats and draws U1 snRNP away from the exon 1 5'ss. They also show that RNA pol II transcription rates influence altered splicing of Htt and that slower RNA pol II transcription, induced by targeted gRNAs and dCas9, favor incomplete splicing and HTT exon 1 production. The manuscript is well written and the results are generally convincing, but I have noted several issues below that should be addressed.

Major points:

1. Fig. 2. To confirm the observed effects of SRSF6 overexpression (OE) are specific to this splicing factor, the OE effect of at least one additional SR protein should be tested.

We have now tested the effects of overexpression of three additional SR proteins: SRSF1, SRSF2 and SRSF3, on the splicing pattern of our minigenes (Supplementary Figure 4). We did not observe the same stimulatory effect on the amount of incomplete splicing, which we had observed when SRSF6 was overexpressed (compare new Fig. 3 with Supplementary Figure 4). On the contrary, the overexpression of all three SR protein led to a decrease in the amount of incompletely spliced minigenes while the canonically spliced transcript levels remained unchanged.

2. Fig. 2D. The higher level of CAG100 incomplete splicing with human vs mouse SRSF6 is attributed to only partial spliceosome integration by the mouse protein. This is a reasonable conclusion since the mouse Srsf6 transcript levels were higher than human but there is no evidence showing that the corresponding mouse and human SRSF6 protein levels are similar.

We have now included data showing SRSF6 protein levels (Fig. 3C). The level of mouse and human SRSF6 overexpression was comparable for both CAG repeat lengths. Furthermore, we describe the protein sequence differences between mouse and human SRSF6 in more detail on page 7 of the current manuscript.

3. Fig. 2E. The surprising finding that human SRSF6 levels were lower in CAG100 vs CAG7 controls (siRNAscrt) should be discussed in more detail.

We thank the reviewer for pointing this out to us. Indeed, when we analysed protein levels after knock-down of SRSF6 (Fig. 3G), we did not observe a difference. We adapted the discussion accordingly to include these new data.

4. Fig. S4. The SDHA and SPP1 minigenes are used to examine CAG-induced mis-splicing but the repeats are only inserted into introns. What is the result if these CAG repeats are inserted into an exon at the same distance from the 5'ss as HTT E1?

We know from human and mouse huntingtin, and from human ataxin 3, that when an expanded CAG repeat is placed in a certain spatial relation to a 5' splice site, that incomplete splicing occurs. We used the SDHA and SPP1 minigenes to test whether the presence of an expanded CAG repeat was

sufficient to cause incomplete splicing, and found this not to be the case. We have edited the text to make this clearer.

5. Discussion. The authors' HD model illustrates SRSF6 binding to the CAG repeat causes misguiding of U1 snRNP to the expansion region versus the E1 5'ss but what is the experimental evidence that supports this CAG-SRSF6-U1snRNP complex formation?

We demonstrated in a previous publication that SRSF6 binds to the huntingtin transcript when it contains an expanded CAG repeat (Sathasivam et al., Proc Natl Acad Sci U S A. 2013 Feb 5; 110(6), 2366-70). The formation of SR protein-U1snRNP complexes, in particular the SR-SR domain interaction of SR proteins and U1-70K, has been shown in several publications as reviewed in e.g. Graveley, RNA. 2000 Sep;6(9):1197-211, Hastings and Krainer, Curr Opin Cell Biol. 2001 Jun;13(3):302-9 and Long and Caceres, Biochem J. 2009 Jan 1;417(1):15-27 (see also references therein).

Minor points:

6. Results P5. The original repeat series was 7, 23, 32, 41, 54 and 103 followed by generation of isogenic 293 lines that I assume resulted in clonal variation in repeat sizes (e.g., 53, 55 in Fig. S1) so for clarity it would be helpful to include a sentence to that effect.

These CAG repeat sizes: 7, 23, 32, 41, 54 and 103 did not arise from clonal variation, but the clones were designed to contain these CAG repeats. The 55CAG in the original Fig. S1 was a contraction from 150CAGs.

7. Fig. S1B. The Q103 fragment appears to be the broader band in the 103 lane but this separation is unconvincing and would be clearer if the electrophoresis run time was increased.

These gels were run so as to contain all potential fragment sizes. The band in question can be seen clearly in the inputs in the updated Supplementary Figure 2D, E and F (F is the same as the new Fig. 1E), where it is not running close to the IgG heavy chain. It is also present for the two Q50 lines, in Fig S1E, where the bands in question are not positive for the FLAG tag.

8. Fig. 2E and P6. The text on P6 indicates that siRNA KDs of both mouse Srsf6 and human SRSF6 resulted in decreased transcript levels but only human is shown in Fig. 2E.

We apologise, the text was incorrect and should have just referred to the human gene. This has been changed in the text.

Reviewer #2 (Remarks to the Author):

This is an important and systematic study providing details of the essential regions of the 5'HTT gene necessary for normal splicing as well as for incomplete splicing, which was first described by this group. This incomplete splicing to create an exon1 protein is a critical new finding in the field that has significant therapeutic implications, including for HTT lowering approaches and alternative targets. Defining the critical regions extends an understanding of how this exon 1 splice variant is produced and the studies provide important insights into the mechanisms involved that can be used to develop models to ascertain whether the production of exon1 protein through alternative splicing is a required step in pathogenesis. Using the mouse mini-gene is a creative approach to investigating mechanisms and the generation of isogenic cell lines expressing these constructs is a critical step to

enable quantitative analysis. The authors also test whether overexpression or decreased SRSF6 modulates incomplete splicing of HTT and of note, there is a selective effect on incomplete splicing (versus canonical splicing) in the context of a highly expanded repeat. It is also intriguing that the position of the CAG repeats modulates the incomplete splicing phenotype. The authors then go on to show that the transcriptional speed of PolII modulates incomplete splicing and evaluate a small molecule that inhibits PolII elongation, as well as introducing adding in tethered CRISPR/Cas complexes that hinder PolII and increased incomplete splicing depending where it is introduced.

1) References missing on p2- 2nd paragraph of intro

We are sorry for this omission and have added the missing references.

2) It would be helpful to have a bit more description of the cell model and rationale for how developed. For instance, although it is clear that the mouse Htt minigenes were used to be able to distinguish between the minigene and endogenous HTT transcripts, it would be helpful to know how conserved the sequences in the constructs are to human HTT, given that the cell line used is human.

We have discussed this in more detail as requested.

3) The initial description of the constructs is a bit confusing, although it becomes more clear in the results as the experiments are described. For general reader, the initial descriptions of the constructs and rationale for each needs to be a bit better clarified. For instance, the relevance of the long intron is not initially clear.

We have realised that Fig. 1, as presented, was very confusing, and this probably explains why the referee felt that the initial description of the constructs was not well explained. We have separated Fig. 1 into two figures and described the rationale for the specific constructs in much more detail.

4) Does the presence of the expanded repeat in the Q100 isogenic line (or other expanded CAG repeat lines) show aggregation or other phenotypes that could be assessed in these assays as well? In particular does treatment with DRB and reduction of incomplete splicing influence aggregation.

We were unable to detect any aggregates in the Q100 lines by immunocytochemistry. This is most likely because these are rapidly dividing cells and the constructs are not expressed under the control of a strong promoter.

Reviewer #3 (Remarks to the Author):

This group previously reported that alternative splicing of Htt gene produces Exon1 peptide (Htt-Exon1) without cleavage by proteases from full-length Htt protein. The discovery revived the significance of previously generated and world-widely used mouse model R6/2. In this new work, the authors investigated the transcriptional speed and skipping exon1-intron1 junction. They also showed the effect of a splicing-related factor SRSF6, which they previously reported as a factor binding to the elongated CAG repeat and affect the alternative splicing, on the skipping of the exon-intron junction. Some of their experiments included beautiful results, while it is still difficult for this reviewer to be convinced about the mechanism. There are two major points that reduces the significance of this paper.

Our previous paper did not really describe alternative splicing or skipping of an exon 1- intron junction, as there was no exon exclusion. Both the small transcript and the full-length transcript

contain exon 1. Our current working model is that the small transcript is generated by a block in the splicing of exon 1 to exon 2.

Major points

1) Some researchers investigated the relationship between transcription speed and alternative splicing. Basically, when the speed is slowed down, the stochastic chance for splicing related factors bind to an exon-intron junction of pre-mRNA is generally increased to up-regulate the splicing efficiency between neighboring exons (Fong et al, Gen Dev 2014; Bentley, Nat Rev Genet 2014). This is a contrastive scheme against the authors' findings in this paper. Naturally there would be an exceptional case that does not follow the general rule.

Our model suggests that SRSF6, by binding to the elongated CAG repeat, results in the sequestration of the splicing complexes, away from the splice donor, to an ectopic location. Slower transcription, could increase this process and allow for the transient interaction to be stabilised. We have included the reviewers comment and have made this clearer by proving a legend to the schematic in Fig 7.

However, the authors should reveal more detailed mechanism, not just showing the dose relationship between SRSF6 and splicing, with more intensive data on the change in splicing machinery under CAG repeat expansion.

On page 9 we have included a reference to a new supplementary figure (Supplementary Figure 5) showing the expression changes in the composition of the splicing machinery in several tissues from an HD mouse model at 6 months of age and HD post mortem BA4 cortex. There is no major dysregulation of the splicing machinery, and there are no consistent changes between tissues. We have previously shown that incomplete splicing occurs in these HD mice at 2 months of age (Sathasivam et al., Proc Natl Acad Sci U S A. 2013 Feb 5; 110(6), 2366-70), prior to the onset of transcriptional dysregulation, and also, in this post mortem brain region (Neueder et al., Sci Rep. 2017 May 2;7(1):1307). Furthermore, we have analysed the effects of overexpression of SRSF1, SRSF2 and SRSF3 on the splicing pattern of the minigenes (new Supplementary Figure 4). We found that the overexpression of these splicing factors did not result in an increase of incomplete splicing, which we had observed by overexpressing SRSF6 (compare new Fig. 3 with Supplementary Figure 4). On the contrary, the overexpression of all three SR protein led to a decrease in the amount of incompletely spliced minigenes while the canonically spliced transcript levels remained unchanged.

2) In this sense, it is not clear how SRSF6 modulates the splicing at the exon-intron junction. The slow transcription would increase the chance of SRSF6 to bind to CAG repeat sequence, but how SRSF6 inhibits the splicing. With direct evidences such as changed components of splicing complex at the Htt exon-intron junction or KD experiments in vitro, the authors should reveal the mechanism how it is related to the skipping.

As mentioned above, we do not really observe a 'skipping' phenomenon, but rather the absence of a splicing event. We demonstrated in a previous publication that SRSF6 binds to a huntingtin transcript when it contains an expanded CAG repeat (Sathasivam et al., Proc Natl Acad Sci U S A. 2013 Feb 5; 110(6), 2366-70). As mentioned above, our model suggests that SRSF6, by binding to the elongated CAG repeat, results in the sequestration of the splicing complexes, away from the splice donor, to an ectopic location. We have added a legend to the schematic in Fig 7 to make this clearer.

Minor points

1) Some of the experiments are difficult to catch what they actually did, and detailed methods should be described. Especially, Fig 1D-F are hard to understand what methods and parameters they actually used for calculation of the ratios and how they evaluate the significance of differences in statistics after reading Fig legend and methods.

We apologise that Fig 1 was difficult to understand. We have now revised these data into two figures with a more detailed explanation.

Reviewer #4 (Remarks to the Author):

The thesis of this study is that transcriptional events affect the translation of an exon1-intron1 mutant huntingtin fragment. The importance of the study is that the mutant fragment could have important consequences on the pathogenesis of HD. This idea is interesting and important. Sections of the manuscript are devoted to the concept that the CAG repeat slows Pol II activity, thereby facilitating the production of the mutant HTT fragment from the exon1-intron1. Experiments are presented that test this concept. Clearly, the studies are involved, requiring invention of cell lines. Over-riding concerns are that the novel approaches need to be confirmed (CRISPR), that small differences (though significant) might not be biologically important, and that changes in mRNA do not necessarily correlate with changes in specific proteins (Westerns or similar). Splicing is complex, so that attribution of the changes to SRSF6 (not part of spliceosome) may be incomplete. Specific points on each Figure are meant to clarify a reader's analysis of data. Reasons to do some of the experiments should be made obvious.

In previous publications, we have shown that all knock-in mouse models, as well as YAC128 mice produce the highly pathogenic exon 1 HTT protein through incomplete splicing (Sathasivam et al., Proc Natl Acad Sci U S A. 2013 Feb 5; 110(6), 2366-70). In all of these mouse models, we have also shown that the exon 1 HTT protein is present by western blotting. We are in the process of conducting experiments that will address the extent to which the exon 1 HTT protein contributes to pathogenesis in knock-in HD mouse models. We have also shown that this incomplete splicing event occurs in fibroblast lines and *post mortem* brains from HD patients (Neueder et al., Sci Rep. 2017 May 2;7(1):1307).

In this paper, we have designed mini-gene systems to provide insights as to mechanisms by which this incomplete splicing might occur. These minigenes systems are not models of disease. They are purely designed to give insights into the mechanisms occur *in vivo* in the context of the 20 kb intron 1 and the complete *Htt* gene.

Although SRSF6 is not part of the snRNA containing core complexes, it is part of the splicing machinery – as indicated in the new Supplementary Figure 5.

Figure 1. The authors made cell lines that express long, medium, short and exon 1 only introns with associated CAG lengths in series. It is supposed that the cell lines, though easier to use and more accessible to experimentation than live tissue, will mimic the natural state of mutant HTT. Shortcomings of the results include (1) lack of detail in figure legend for D-F. In Fig 1F, is the curve (not clear how it is fit) related to changes in SFSR6, as shown in Figure 2. Northern blots would more clear show the splice isoform than would the PCR products. Figure 1D requires more detail. What is the point of the figure?

We apologise that Fig 1 was difficult to understand. We have now revised these data into two figures with a more detailed explanation. Because of our lack of clarity, the reviewer has mistaken our 3'RACE gels for PCR products. Given this, we understand why he/she has asked to be able to visualise the transcripts on a northern blot. However, we have previously used RNAseq show the identity of the splice isoform (Sathasivam et al., Proc Natl Acad Sci U S A. 2013 Feb 5; 110(6), 2366-70). In this paper, all of the comparative transcript levels were quantified by qPCR and we have now provided a schematic to illustrate the location of these all of these assays (new Fig. 2B).

page 5, bottom. The sentence about incompletely spliced transcripts and endonucleolytically cleaved pre-mRNA is not experimentally demonstrated.

The 3'RACE demonstrates that these polyadenylation sites are activated and polyadenylation is an endonuclease event.

page 6 at top. Supplementary Figure 1 should be a figure in the main paper.

We have transferred one of the panels to the new version of Fig. 1E.

page 6. The last part of the last paragraph is confusing. Why is the SRSF6 lower in the CAG100 cells? This finding is potentially important to interpret the data. Are there other changes in the cells that might account for different results between the CAG 100 and CAG7 cells? In all, this paragraph is confusing. It is unclear if SFSR6 is in the spliceosome, so that the term "not fully integrate into the human spliceosome" might be misleading.

We have now included data showing SRSF6 protein levels after knock-down (Fig. 3G). We did not observe a corresponding difference in the levels of SRSF6 protein between the CAG7 and CAG100 cell lines, to that identified at the transcript level (Fig. 3F). Furthermore, we describe the protein sequence differences between mouse and human SRSF6 in more detail on page 7 of the current manuscript. We have adapted the discussion accordingly to include these new data.

Figure 2. Compare this figure to Figure 1F. There could be variability in the SFSR6 abundance in the cells. It would be important (critical) to measure the amount of SFSR6 protein. mRNA does not always translate to protein. In knockdown studies, the Western would help to interpret the extent of change of the SFSR6 protein. Furthermore, splicing factors can be autoregulated, so that Westerns become imperative. [...] In 2E, the difference between mRNA CAG7 and CAG100 should be confirmed by Western blot.

In Fig. 3C we now show SRSF6 protein levels after overexpression. The level of mouse and human SRSF6 overexpression was comparable for both CAG repeat lengths.

In Fig. 3G we now show SRSF6 protein levels after siRNA mediated knockdown. As mentioned above, we did not observe the difference in basal SRSF6 levels, which we had seen on transcript level. However, for both CAG repeat lengths we observed a statistically significant down-regulation of SRSF6 protein levels after siRNA treatment by 50% and 40%, respectively.

There is no binding data shown, so that the last paragraph on page 7 is conjecture.

We have previously shown the increased binding of SRSF6 to the elongated CAG repeat (Sathasivam et al., Proc Natl Acad Sci U S A. 2013 Feb 5; 110(6), 2366-70).

Any splicing data should be confirmed by Northern blot, better suited to demonstrate splicing.

As mentioned above, our RNAseq data confirmed the nature of the incompletely spliced transcript (Sathasivam et al., Proc Natl Acad Sci U S A. 2013 Feb 5; 110(6), 2366-70).

In general, there are small differences in many of these plots -- not clear if the small differences are biologically important.

Our minigenes systems are not models of disease, but rather as minimalistic proof of concept systems. Our experiments are meant to provide insights as to mechanisms by which incomplete splicing of the *Htt* gene might occur. Differences between the CAG repeat lengths provide insights into these underlying mechanisms and should not be seen as potential therapeutic targets.

Page 8 and Figure 3. This study tested the idea that placement of CAG repeats could affect splicing. A compelling reason for this study is not presented. What is the real impact of the experiment? The reader ought to know why the experiment has a bearing on the HD pathogenesis.

We apologies and have explained this more clearly on page 9 and 10.

Figure 4 studies the effect of the rate of transcription of Pol II on the amount of incomplete splicing. A good idea. However, the data show a series of steady state measurements, rather than a real rate. The interpretation will be indirect. In Figure 4A, the curve fit should be clarified. The use of "regression fits" should be clarified.

We agree that we are inferring the rate of transcription from Pol II occupancy. We have included a supplementary table (Supplementary Table 3) with regression fit statistics and described the regression fits in the materials and methods statistics section.

Figure 5. The CRISPR/Cas9 studies test the idea that slowing transcription presents "obstacles" to increase the incomplete splicing. The figures actually show little change (although significant) and raises the concern that the changes might not be biologically important. Would a 10% change in the mRNA really change production of a mRNA, or protein, in a meaningful way? The use of CRISPR in this way would need to be better characterized as presenting an obstacle. Indeed, on page 11 it is said the "we hypothesized that these tethered CRISPR/Cas complexes would hinder Pol II transcription..". The hypothesis would need to be secured for the experiment to be interpretable.

As we stated above the minigenes are not models of disease. The binding of CRISPR/dCas9 to DNA is non-covalent and is transient, and we would not expect a complete block in transcription. We thank the referee for pointing out that our text could be misinterpreted, and we have edited this. The proof of concept, that CRISPR/Cas9 complexes represent obstacles for transcription can be seen in Fig. 5B (now Fig. 6B). This shows that tethering of these complexes to a very 5' region (5' UTR) resulted in reduction of overall minigene transcript levels, while splicing patterns were not changed.

The discussion is difficult to interpret based on the data as presented.

We hope that the revisions to the manuscript make the discussion easier to interpret.

Reviewers' comments:

Reviewer #1 (Remarks to the Author):

This revision addresses all of my prior issues.

Reviewer #2 (Remarks to the Author):

The reviewer concerns have been addressed

Reviewer #3 (Remarks to the Author):

1. The authors claim "Our previous paper did not really describe alternative splicing or skipping of an exon 1- intron junction, as there was no exon exclusion. Both the small transcript and the full-length transcript contain exon 1. Our current working model is that the small transcript is generated by a block in the splicing of exon 1 to exon 2."

The comment in rebuttal letter might come from different definition of alternative splicing or meaning of skipping, but it is not a problem to influence the evaluation.

The explanation of the authors about transcriptional speed and splicing efficiency is still difficult to understand. SRSF6 binds to CAG in genome DNA but not transcripts in their model. The amount of SRSF6 sequestered to CAG would not differ in slow and fast transcription. If the RNA transcripts stay for a longer time, there will be more chance for the residual (non-sequestered) SRSF6 to be recruited to the U1 splicing complex.

In this regard, the local amount of SRSF6 at the Htt transcript would be more important rather than the transcription speed. However, the original study of Prof Krainer by the established RNA analysis (EMBOJ 1995) showed the increase of SRSF6/SRp55 did not significantly change the ratio of spliced and unspliced transcripts.

Another question is whether SRSF6 sequestered to CAG repeat is functional. DNA-ChIP together with RNA-ChIP assays is necessary to show the balance of DNA-attached SRSF6 and RNA recruited SRSF6. Also the U1 complex at exon-intron junction needs to be quantified by ChIP in their model.

2. Supplementary Figure 5 is interesting but seems difficult to integrate into the hypothesis. The authors assume sequestration of SRSF6 and local hyper-function of this molecule to suppress splicing of exon1-intron1. It would be necessary to confirm ectopic distribution of SRSF6 does not affect splicing of other genes, rather than to examine that splicing factors by themselves are not affected in HD.

This manuscript has been improved substantially by revision, but still some main concerns remain and further investigation is essential for judgment.

Reviewer #4 (Remarks to the Author):

This manuscript is substantially revised. In particular, Figure 1 (which confused all of the reviewers) has been revised as two figures. Much better.

The CRISPR study is described as a minimalistic approach to mechanism. Not sure what that means.

The rate of transcription is really a stream of individual time points, and should be clarified as such.

Overall, a much better organization and exposition of worthy studies.

Reviewer #1 (Remarks to the Author):

This revision addresses all of my prior issues.

We thank the reviewer for her/his helpful comments during the review process.

Reviewer #2 (Remarks to the Author):

The reviewer concerns have been addressed

We thank the reviewer for her/his helpful comments during the review process.

Reviewer #3 (Remarks to the Author):

1. The authors claim “Our previous paper did not really describe alternative splicing or skipping of an exon 1- intron junction, as there was no exon exclusion. Both the small transcript and the full-length transcript contain exon 1. Our current working model is that the small transcript is generated by a block in the splicing of exon 1 to exon 2.”

The comment in rebuttal letter might come from different definition of alternative splicing or meaning of skipping, but it is not a problem to influence the evaluation.

The explanation of the authors about transcriptional speed and splicing efficiency is still difficult to understand. SRSF6 binds to CAG in genome DNA but not transcripts in their model.

SRSF6 does not bind to genomic DNA in our model. We are not sure why the referee thinks that this is the case, as we do not state this anywhere in the paper. To our knowledge splicing factors do not bind to DNA.

The amount of SRSF6 sequestered to CAG would not differ in slow and fast transcription.

We do not claim that SRSF6 has to have a direct effect on transcription speed. In our model, there is no requirement for it to do this (see Figure 7).

If the RNA transcripts stay for a longer time, there will be more chance for the residual (non-sequestered) SRSF6 to be recruited to the U1 splicing complex. In this regard, the local amount of SRSF6 at the *Htt* transcript would be more important rather than the transcription speed.

In our model, SRSF6 recruits the U1 complex to an ectopic location by binding to the CAG repeat in the transcript. We propose that this together with the slowing of transcription in the region of the microsatellite contributes to the incomplete splicing event. We propose that these are dynamic processes regulated by binding and enzyme kinetics.

However, the original study of Prof Krainer by the established RNA analysis (EMBOj 1995) showed the increase of SRSF6/SRp55 did not significantly change the ratio of spliced and unspliced transcripts.

In this paper, Krainer et al. only studied two transcripts (human beta-globin and adenovirus E1A gene) in a cell line overexpressing both of them and admitted the limitations of their study in the discussion. However, we are not suggesting that SRSF6 has a global effect on splicing in the presence of the HD mutation. Instead, we are focussing on its effect on the splicing of the *Htt* gene. Krainer et al. did not study the *Htt* gene in the context of a CAG repeat mutation.

Another question is whether SRSF6 sequestered to CAG repeat is functional. DNA-ChIP together with RNA-ChIP assays is necessary to show the balance of DNA-attached SRSF6 and RNA recruited SRSF6.

We have shown that we can modulate *Htt* splicing by changing the levels of SRSF6, therefore, SRSF6 levels have functional consequences. Performing DNA-ChIP does not make sense for an RNA binding protein. Similarly, RNA-chromatin immunoprecipitation (ChIP) does not make sense. We have no reason to think that DNA-attached SRSF6 exists.

However, we have previously published that we can immunoprecipitate *Htt* RNA with an antibody to SRSF6 (Sathasivam et al. 2013 PNAS 110, 2366-2370).

Also the U1 complex at exon-intron junction needs to be quantified by ChIP in their model.

Chromatin immunoprecipitation to look at U1 binding to packaged DNA, does not make sense.

2. Supplementary Figure 5 is interesting but seems difficult to integrate into the hypothesis. The authors assume sequestration of SRSF6 and local hyper-function of this molecule to suppress splicing of exon1-intron1. It would be necessary to confirm ectopic distribution of SRSF6 does not affect splicing of other genes, rather than to examine that splicing factors by themselves are not affected in HD.

Supplementary Figure 5 was provided in response to one of the other referees, who questioned whether the changes in *Htt* splicing could be attributed to SRSF6, rather than other splicing factors. In order to respond to the reviewer, we would need to be able to demonstrate that other genes are not alternatively spliced in HD mouse models at young presymptomatic ages, when we can detect incomplete splicing. As we state in the paper, there are RNAseq data available that show that HD-related transcriptional dysregulation does not occur at this early time point. However, these RNAseq datasets do not have sufficient read-depth to answer this question related to alternative splicing. We cannot exclude that fact that the ectopic location of SRSF6 might affect the splicing of other genes, but at the moment do not have the resources to answer this question.

This manuscript has been improved substantially by revision, but still some main concerns remain and further investigation is essential for judgment.

Reviewer #4 (Remarks to the Author):

This manuscript is substantially revised. In particular, Figure 1 (which confused all of the reviewers) has been revised as two figures. Much better.

Thank you, we apologise for the original version of Fig. 1.

The CRISPR study is described as a minimalistic approach to mechanism. Not sure what that means.

We made this statement in our response to the reviewers, and not in the paper. We meant that we had used a model with the essential elements for splicing, and this was not a model of disease.

The rate of transcription is really a stream of individual time points, and should be clarified as such.

We agree with the reviewer. We had made the following statement in the 'response to reviewers', we have now added a sentence to the text on page 11, to clarify that we are inferring transcription speed from occupancy rates.

'We agree that we are inferring the rate of transcription from Pol II occupancy. We have included a supplementary table (Supplementary Table 3) with regression fit statistics and described the regression fits in the materials and methods statistics section.'

Overall, a much better organization and exposition of worthy studies.

We thank the reviewer for her/his helpful comments during the review process.

REVIEWERS' COMMENTS:

Reviewer #3 (Remarks to the Author):

The authors logically addressed previous concerns, and this reviewer finds the paper is acceptable. (Fig 7 is a little bit confusing, because they show cartoon of Pol II, which makes readers to imagine DNA, just below the RNA transcript. The authors should better clarify that it is RNA transcripts, if possible).

Reviewer #4 (Remarks to the Author):

This reviewer is now satisfied with the revised manuscript.

REVIEWERS' COMMENTS:

Reviewer #3 (Remarks to the Author):

The authors logically addressed previous concerns, and this reviewer finds the paper is acceptable. (Fig 7 is a little bit confusing, because they show cartoon of Pol II, which makes readers to imagine DNA, just below the RNA transcript. The authors should better clarify that it is RNA transcripts, if possible)

We would like to thank the reviewer for her/his advice on readability of Fig. 7 and have changed the figure accordingly.

Reviewer #4 (Remarks to the Author):

This reviewer is now satisfied with the revised manuscript.

We thank the reviewer for her/his helpful comments during the review process.